# See&Trek: Training-Free Spatial Prompting for Multimodal Large Language Model

**Pengteng Li**
AI Thrust, HKUST(GZ)
AI$^2$ROBOTICS

**Pinhao Song**
KU Leuven

**Wuyang Li**
EPFL

**Huizai Yao**
AI Thrust, HKUST(GZ)

**Weiyu Guo**
AI Thrust, HKUST(GZ)
AI$^2$ROBOTICS

**Yijie Xu**
AI Thrust, HKUST(GZ)

**Dugang Liu**
SZU

**Hui Xiong**[*]
AI Thrust, HKUST(GZ)
CSE, HKUST

## Abstract

We introduce **SEE&TREK**, the first training-free prompting framework tailored to enhance the spatial understanding of Multimodal Large Language Models (MLLMS) under vision-only constraints. While prior efforts have incorporated modalities like depth or point clouds to improve spatial reasoning, purely visual-spatial understanding remains underexplored. **SEE&TREK** addresses this gap by focusing on two core principles: increasing visual diversity and motion reconstruction. For visual diversity, we conduct Maximum Semantic Richness Sampling, which employs an off-the-shell perception model to extract semantically rich keyframes that capture scene structure. For motion reconstruction, we simulate visual trajectories and encode relative spatial positions into keyframes to preserve both spatial relations and temporal coherence. Our method is training&GPU-free, requiring only a single forward pass, and can be seamlessly integrated into existing MLLMS. Extensive experiments on the VSI-BENCH and STI-BENCH show that **SEE&TREK** consistently boosts various MLLMS performance across diverse spatial reasoning tasks with the most +3.5% improvement, offering a promising path toward stronger spatial intelligence. The link of code: `https://github.com/Hoantrbl/SeeTrek`.

## 1 Introduction

Multimodal Large Language Models (MLLMS) [1, 2, 3, 4] have witnessed rapid advancements, demonstrating impressive capabilities in understanding and generating cross-modal content. By integrating visual and textual inputs, these models have shown potential in various tasks such as image captioning [5], visual question answering [6], and embodied AI [7, 8, 9].

Spatial reasoning is crucial for empowering MLLMs to understand and interact with real-world environments, particularly in tasks involving object localization, motion prediction, and physical interactions. Enhancing MLLMs with spatial awareness can significantly improve their performance on downstream applications such as navigation [10] and robotic manipulation [8, 11]. Recent works have introduced depth cues [3], camera poses [12], and 3D priors [13] into MLLM training, aiming to construct more comprehensive spatial representations [14, 15]. However, despite these advances, existing models still struggle to robustly reason about complex spatial relationships, particularly in scenarios requiring multi-step reasoning, complex visual and temporal dynamics, or generalization to novel environments. *We reflect critically on two pivotal factors that hinder current* MLLMS *from overcoming the spatial understanding bottleneck:*

---

[*]Corresponding author.

39th Conference on Neural Information Processing Systems (NeurIPS 2025).

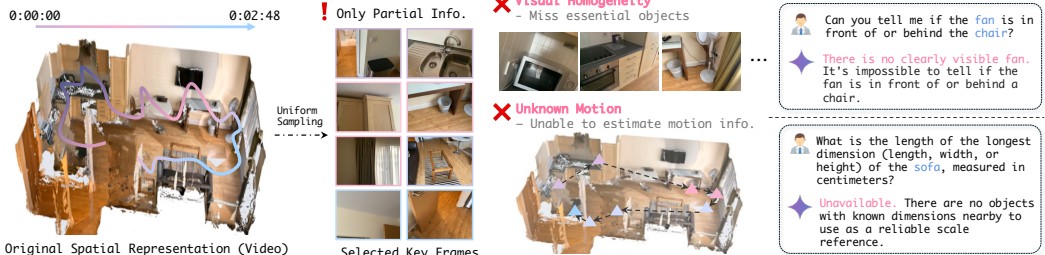

Figure 1: Illustration of insufficient spatial understanding ability of current MLLMs. They adopt uniform sampling to extract the video with spatial semantics, leading to visual homogeneity and unknown motion. SEE&TREK aims to tackle these two problem, which can be referred to Figure 3.

**1) Visual Homogeneity**: As shown in Figure 1, generally, a large number of existing MLLM pipelines adopt uniform temporal sampling strategies, in which 8 or 32 frames are used as input due to limited GPU memory when selecting keyframes from the video which captures full spatial semantics. Without any structural constraints or prior knowledge, this uniform temporal sampling often captures the frames (i) without any salient features (e.g., walls, ceilings, and floors) (ii) or containing fragments of objects. Those frames will decrease the signal-to-noise ratio in the input frames, limiting the MLLM's ability to reconstruct or reason about the full spatial layout.

**2) Unknown Motion**: Relying solely on sampled frames, without access to explicit ego-motion information, significantly impairs a model's ability to infer object movement and displacement within a scene. Such capabilities are critical to spatial reasoning tasks, including estimating object distances, predicting motion trajectories, and establishing temporal order. In the absence of explicit motion cues, MLLMs are forced to rely primarily on commonsense priors acquired during pretraining, rather than on directly grounded visual evidence. Consequently, the spatial predictions made by such models tend to be speculative rather than evidence-based, revealing a fundamental limitation of existing vision-only MLLM architectures.

To tackle the two above-mentioned important factors, we propose **SEE&TREK**, a simple yet effective training&GPU-free spatial prompt method to jointly boost spatial-temporal reasoning in MLLMs. *To tackle the visual homogeneity,* we utilize an off-the-shelf perception model, e.g., object detectors, to extract semantically rich keyframes that capture the spatial structure of scenes to increase visual diversity. *To tackle unknown motion,* we leverage Visual Odometry (VO) to simulate visual trajectories from the given videos and add extra motion cues into the keyframes, preserving spatial relationships and temporal coherence. Featured with 1) training&GPU-free, 2) plug-and-play, and 3) single-forward characteristics, **SEE&TREK** can be seamlessly integrated with open-source MLLMs or commercial engines. Comprehensive experiments on VSI-BENCH and STI-BENCH verify that **SEE&TREK** significantly improves the performance of multiple MLLMs across diverse spatial reasoning tasks, offering a promising direction for enhancing the spatial understanding of MLLMs. In summary, our contributions are as follows:

- We introduce **SEE&TREK**, the first **training- and GPU-free** spatial prompting framework to enhance the spatial understanding capabilities of MLLMs. It is **plug-and-play**, **single-forward**, and compatible with both open-source and commercial MLLMs.
- To provide rich spatial and motion cues, we design a comprehensive spatial prompting strategy: 1) Maximum Semantic Richness Sampling: We use off-the-shelf perception models (*e.g.*, YOLO) to extract semantically rich and diverse keyframes from videos. 2) Motion Reconstruction: We reconstruct camera motion in BEV and 3D using Visual Odometry and label keyframes with motion cues to preserve spatial and temporal coherence.
- Comprehensive experiments on VSI-BENCH and STI-BENCH show that **SEE&TREK** significantly boosts the spatial understanding performance of multiple MLLMs across diverse scales and architectures, offering a promising direction for future spatial intelligence.

## 2 Related Works

**Long Video Understanding.** Most evaluations of spatial understanding in MLLMs focus on long video comprehension [5, 16], where models analyze dynamic scenes and answer related questions. A key challenge in both spatial and video understanding is maximizing relevant information retention.

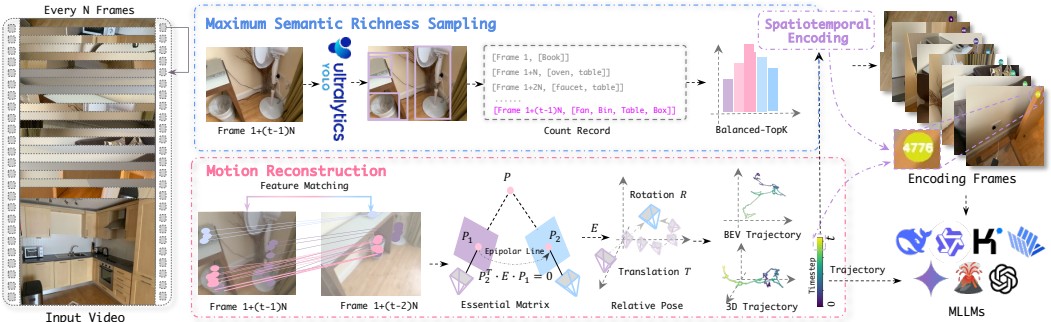

Figure 2: Overview of the proposed **SEE&TREK**. We first sample one frame for every $N$ frame for post-processing. For Maximum Semantic Richness Sampling, a visual perception model *e.g.* YOLO [40] detects objects in each sampled frame, updating the category count. To boost scene representation, we propose a *Balanced-TopK* strategy for selecting semantically rich frames. For Motion Reconstruction, we perform feature matching with the previous frame. Then, matching points are used to estimate the essential matrix $E$ via RANSAC [41], from which the relative camera pose and motion trajectory are recovered. Finally, SPATIOTEMPORAL ENCODING is introduced to integrate motion and spatial cues for a more comprehensive representation.

Existing methods often employ specialized adapters [17, 18], tokenization strategies [19, 20], or memory modules [21, 22, 23] to compress and store video semantics. Other approaches convert videos into document-style inputs [24], use retrieval-augmented generation (RAG) frameworks [25, 26, 27, 28], or structure content in trees [28] to reduce memory load. However, these techniques generally require fine-tuning MLLMs or rely on VLMs for information retrieval, which is both computationally expensive and resource-intensive. Recent work has explored query-based frame selection [29, 30, 31, 32, 33], combining traditional heuristics with feedback from pretrained VLMs. Building on these insights, we propose a novel GPU-free keyframe selection strategy that identifies representative frames efficiently without additional training or heavy computational demands.

**Spatial Understanding.** Spatial understanding is essential for deploying MLLMs in real-world applications such as Embodied AI [8, 11] and Autonomous Driving [34, 10]. Recent research has explored leveraging multimodal inputs, such as depth maps or point clouds, to enable explicit [35] or implicit [12, 36, 13] 3D scene modeling, which is processed by the LLM decoder to enhance spatial comprehension. However, these approaches often rely on precise cross-modal alignment and highly customized pipelines, posing challenges for deployment and limiting generalizability across diverse real-world settings. Additionally, common benchmarks like ScanQA [14] and SQA3D [15] typically assume uniform camera motion, making them less suitable for complex, dynamic environments. In contrast, vision-only MLLMs offers greater usability [1, 2, 4] but often struggles with nuanced spatial understanding in video-based tasks [37, 38, 39]. These limitations motivate the development of a spatial prompting strategy that enhances the spatial reasoning capabilities of existing MLLMs.

## 3 Method

As shown in Figure 2, the proposed **SEE&TREK** aims to overcome the limitation of visual homogeneity and unknown motion in the captured keyframes. In detailed, given the video sequence $V = \{f_t\}_{t=1}^{N_v}$ with $N_v$ frames, we sample one frame every $N$ frames of $V$. We first select the rich semantics frames by leveraging an efficient visual perception model and the proposed *Balanced-TopK* strategy and use OV to simulate the camera relative motion trajectory to obtain the moving information. Finally, SPATIOTEMPORAL ENCODING is introduced to encode each frame with color-specified and number marks, which efficiently integrates both important properties. The algorithm is presented in Algorithm 1 and the detailed version can be found in Algorithm A.1 in the appendix.

### 3.1 Maximum Semantic Richness Sampling

To address the limitation of **visual homogeneity**, the first step of **SEE&TREK** is to select the keyframes that convey the richest semantic content to increase visual diversity. Since inference resources constrain us to only a few frames, we posit that the frame containing the most detected objects will best capture the scene's diversity. Concretely, we run a pretrained YOLO detector $\mathcal{Y}(\cdot)$

**Algorithm 1** SEE&TREK: Maximum Semantic Richness Sampling & Motion Reconstruction

---

**Require:** Video $V = \{f_t\}_1^{N_v}$, interval $N$, #keyframes $K$, detector $\mathcal{Y}$, intrinsics $\mathbf{K}$
**Ensure:** Keyframes $\{\mathcal{F}'_{t_i}\}_{i=1}^K$, $\mathbf{P}_{\text{BEV}}$, $\mathbf{P}_{\text{3D}}$, $\mathcal{T}^{\text{world}}$
 1: $\mathcal{S} \leftarrow \{f_t \mid t \bmod N = 0\}$                                          // Uniform Subsampling
 2: $\mathcal{T}^{\text{world}} \leftarrow [], \mathcal{O} \leftarrow \emptyset$
 3: **for** each pair $(f_{t-1}, f_t) \in \mathcal{S}$ **do**
 4:     $(\mathbf{R}_t, \mathbf{T}_t, \mathcal{C}_t) \leftarrow \text{FRAMEPIPELINE}(f_{t-1}, f_t)$          // ORB→Match→RANSAC→Decompose & Perception $\mathcal{Y}$
 5:         $\mathcal{T}^{\text{world}} += \mathbf{T}_t^{\text{world}}, \ \mathcal{O} += (t, \mathcal{C}_t)$
 6: **end for**
 7: $\mathcal{O}^* \leftarrow \text{TRIMSPAN}(\mathcal{O})$                                       // Valid Interval $[t_s, t_e]$
 8: $(\mathcal{T}_{\text{sel}}, \mathcal{C}_{\text{sel}}) \leftarrow \text{SELECTKEYFRAMES}(\mathcal{O}^*, K)$              // Balanced-Top$K$
 9: **for** $t_i \in \mathcal{T}_{\text{sel}}$ **do**
10:     $\mathcal{F}'_{t_i} \leftarrow \text{OVERLAY}(f_{t_i}, \text{ColorMap}(t_i/K))$          // SPATIOTEMPORAL ENCODING: Index & Hue Mark
11: **end for**
12: $(\mathbf{P}_{\text{BEV}}, \mathbf{P}_{\text{3D}}) \leftarrow \text{RENDERTRAJ}(\mathcal{T}^{\text{world}})$                       // ProjectXY + Render3D

---

[40] on every frame $f_t$, yielding a detected class set $\mathcal{C}_t = \mathcal{Y}(f_t) = \{c_1, c_2, \ldots, c_{n_t}\}$, where $n_t$ is the number of detected classes, and each $c_i$ belongs to the predefined label vocabulary $\mathcal{L}$. Thus, within a valid temporal interval $[t_s, t_e]$, we collect a set of detected class set for each frame $f_t$ $\mathcal{C}^* = \{\mathcal{C}_t \mid t_s \leq t \leq t_e\}$.

A simple frame selection method is to choose the Top$K$ frames with the most detected classes. However, this can bias selection toward the temporal interval in which the greatest number of objects appear, causing MLLMs to miss other semantically important parts of the video. To address this, we propose a frame sampling strategy called *Balanced-TopK*, which extracts a set of $K$ keyframes that are both object-rich and temporally diverse. In detail, we first select the initial keyframe $f_{\tau_0}$ as the one with the maximum number of detected classes, where $\tau_0$ denotes the selected keyframe index:

$$\tau_0 = \arg \max_{t_s \leq t \leq t_e} \left( |\mathcal{C}_t| \right), \tag{1}$$

with ties broken by choosing the earliest frame. The initial class pool is set as $\mathcal{C}_{\text{sel}} = \mathcal{C}_{t_0}$. To avoid local bias, ensure temporal uniformity, and capture a broader semantic representation, we partition $\mathcal{C}^*$ into $K-1$ contiguous temporal segments $\{\mathcal{C}^{(1)}, \ldots, \mathcal{C}^{(K-1)}\}$. In each segment $\mathcal{C}^{(k)}$, to increase the overall information gain per frame, we select one frame $\tau_k$ such that the overlap of its detected categories with $\mathcal{C}_{\text{sel}}$ is minimized, classes count is maximized, and the frame is as early as possible:

$$\tau_k = \arg \min_{t \in \mathcal{T}^{(k)}} \left( |\mathcal{C}_t \cap \mathcal{C}_{\text{sel}}|, \ -|\mathcal{C}_t|, \ t \right), \tag{2}$$

where $\mathcal{T}^{(k)}$ is the subset of timesteps, in which each timestep $\tau$ corresponds to a detected class set $\mathcal{C}_\tau$ in $\mathcal{C}^{(k)}$. After each selection, the accumulated category set is updated as $\mathcal{C}_{\text{sel}} \leftarrow \mathcal{C}_{\text{sel}} \cup \mathcal{C}_{\tau_k}$. This process yields a set of $K$ selected keyframes $\{f_{\tau_i}\}_{i=0}^{K-1}$ that are semantically representative, object-abundant, and temporally distributed, forming an optimal subset for downstream understanding.

## 3.2 Motion Reconstruction

As mentioned in the introduction, MLLMs' spatial understanding is limited by **unknown camera motion**, which leads to confusion about movement patterns, object displacement, and the spatial layout of a scene. In this section, we aim to estimate the camera motion via visual odometry (VO) from a monocular video and connect the estimated camera motion with the selected keyframes by painting motion cues on them. Through feeding MLLMs with visualized camera motion and keyframes with motion cues, the spatial understanding capability can be enhanced.

**Camera motion estimation.** We adopt a feature-based VO pipeline utilizing ORB (Oriented FAST and Rotated BRIEF) features and essential matrix estimation [42, 43]. In detail, the current frame $f_t$ and the last frame $f_{t-1}$ are converted to grayscale. We first extract ORB key points and descriptors from each frame as follows:

$$\begin{cases} \mathcal{K}_{t-1}, \mathcal{D}_{t-1} & = \text{ORB}(f_{t-1}), \\ \mathcal{K}_t, \mathcal{D}_t & = \text{ORB}(f_t), \end{cases} \tag{3}$$

where $\mathcal{K}_{t-1} = \{\mathbf{p}_i^{t-1}\}_{i=1}^N$ and $\mathcal{K}_t = \{\mathbf{p}_i^t\}_{i=1}^N$ denote the detected 2D keypoints (in pixel coordinates), and $\mathcal{D}$ represents the corresponding ORB binary descriptors. Feature correspondences are established

by matching descriptors between the two frames, resulting in a set of matched keypoint pairs $\mathcal{M}_{t-1,t} = \left\{ (\mathbf{x}_i^{t-1}, \mathbf{x}_i^t) \right\}_{i=1}^P$, where $\mathbf{x}_i^{t-1}, \mathbf{x}_i^t \in \mathbb{R}^2$ are the matched 2D keypoints in pixel coordinates, and $P$ is the number of matches. To compute the essential matrix, we first normalize the image coordinates using the camera intrinsic matrix $\mathbf{K}$: $\hat{\mathbf{x}}_i = \mathbf{K}^{-1} \begin{bmatrix} \mathbf{x}_i \\ 1 \end{bmatrix}$. The essential matrix $\mathbf{E} \in \mathbb{R}^{3\times 3}$ encapsulates the relative motion between the two frames (up to scale) and is estimated by minimizing the geometric error of the epipolar constraint:

$$\mathbf{E} = \arg\min_{\mathbf{E}'} \sum_i \rho\left( (\hat{\mathbf{x}}_i^t)^\top \mathbf{E}' \hat{\mathbf{x}}_i^{t-1} \right), \tag{4}$$

where $\rho(\cdot)$ is a robust loss function such as the truncated quadratic or Tukey biweight [44], and RANSAC is used to handle outliers [41]. Once the essential matrix $\mathbf{E}$ is estimated, we recover the relative rotation $\mathbf{R}_t \in SO(3)$ and translation direction $\mathbf{T}_t \in \mathbb{R}^3$ (up to scale) by decomposing $\mathbf{E}$ via Singular Value Decomposition (SVD) and applying the Cheirality condition to resolve ambiguity. This results in the relative pose $(\mathbf{R}_t, \mathbf{T}_t)$ between frames $t-1$ and $t$. Assuming the global camera pose at time $t-1$ is known as $(\mathbf{R}_{t-1}^{\text{world}}, \mathbf{T}_{t-1}^{\text{world}})$, the global pose at time $t$ is updated recursively:

$$\begin{cases} \mathbf{R}_t^{\text{world}} = \mathbf{R}_{t-1}^{\text{world}} \mathbf{R}_t, \\ \mathbf{T}_t^{\text{world}} = \mathbf{R}_{t-1}^{\text{world}} \mathbf{T}_t + \mathbf{T}_{t-1}^{\text{world}}. \end{cases} \tag{5}$$

This process is repeated over the entire sampled frame sequence to reconstruct the full camera trajectory. By estimating relative camera motion, we obtain the spatial position of each frame and the overall structure of the scene, thereby offering an implicit spatial reference for the orientation and placement of objects within the space. After obtaining the 3D camera trajectory $\{\mathbf{T}_t^{\text{world}}\}_{t=1}^K$, we generate two visualizations: a top-down Bird's Eye View (BEV) and a 3D trajectory plot, denoted as $\mathbf{P}_{\text{BEV}}$ and $\mathbf{P}_{\text{3D}}$. For visualization, each trajectory point at timestamp $t$ is assigned a color using $\Phi(t) = \text{ColorMap}(t/K)$, where $K$ denotes the total number of trajectory points. The BEV is created by projecting the 3D camera positions onto the XY plane, while the 3D plot retains the full spatial geometry. These visualizations provide a concise and intuitive summary of the camera motion throughout the video.

**Spatiotemporal Encoding.** Though the semantically rich keyframes and camera motion are obtained, MLLMs have difficulty in connecting both information if they are simply fed into MLLMs: *MLLMs have no idea of the camera poses of the corresponding keyframes.* This disconnection hinders the model's ability to understand spatial relationships, track object transitions, or infer occlusion and continuity, particularly in long-range, egocentric sequences. To address this issue, we propose SPATIOTEMPORAL ENCODING by explicitly encoding the motion information into the keyframes. Specifically, two trajectory-aware markers are integrated into each selected keyframe: 1) A frame index representing its position within the motion sequence. 2) A color-coded marker derived from a continuous colormap, indicating temporal progression along the 3D camera motion trajectory. These visual markers are directly overlaid on the image, resulting in augmented frames where each carries both semantic richness (via object presence) and spatiotemporal context (via visual cues).

In detail, given the corresponding camera trajectory and the key-frame indices $\mathcal{T}_{\text{sel}} = \{\tau_i\}_{i=0}^{K-1}$ from Section 3.1, we assign each keyframe $\tau_i \in \mathcal{T}_{\text{sel}}$ a unique RGB color $\mathbf{c}_i$ from the same colormap as the original trajectory, i.e., $\mathbf{c}_i = \Phi(t_i) = \text{ColorMap}(\frac{t_i}{K})$. This ensures a smooth color gradient that reflects the temporal ordering and spatial progression along the motion path. Each frame $f_{\tau_i}$ is then augmented with a simple inpainting operation: we draw a filled circle at a fixed top-right position with color $\mathbf{c}_i$ and overlay the frame index $t_i$ as a label inside the marker (as shown in the bottom-right corner of Figure 2). The resulting modified frame $f'_{\tau_i}$ compactly encodes both the temporal sequence and spatial trajectory through these visual cues.

This transformation preserves the integrity of selected frames while adding motion cues, enhancing multimodal models' ability to associate frames with motion progression and spatial layout. Unlike implicit positional encoding or post-hoc temporal reasoning, our approach offers a direct, efficient, and interpretable way to inject trajectory awareness into the input space. Consequently, MLLMs gain a deeper understanding of scene evolution and the spatial distribution of visual content.

Table 1: Comparison of various MLLMs boosting by **SEE&TREK** on the VSI-BENCH benchmark. † indicates results on VSI-BENCH (tiny) set. * indicates we use the instruct version.

| Methods | Avg. | Obj. Count | Abs. Dist. | Obj. Size | Room Size | Rel. Dist. | Rel. Dir. | Route Plan | Appr. Order |
|---|---|---|---|---|---|---|---|---|---|
| | | | **Numerical Answer** | | | | **Multiple-Choice Answer** | | |
| *Baseline* | | | | | | | | | |
| CHANCE LEVEL (RANDOM) | - | - | - | - | - | 25.0 | 36.1 | 28.3 | 25.0 |
| CHANCE LEVEL (FREQUENCY) | 34.0 | 62.1 | 32.0 | 29.9 | 33.1 | 25.1 | 47.9 | 28.4 | 25.2 |
| *VSI-Bench (tiny) Perf.* | | | | | | | | | |
| †HUMAN LEVEL | 79.2 | 94.3 | 47.0 | 60.4 | 45.9 | 94.7 | 95.8 | 95.8 | 100.0 |
| †GEMINI-1.5 FLASH | 45.7 | 50.8 | 33.6 | 56.5 | 45.2 | 48.0 | 39.8 | 32.7 | 59.2 |
| †GEMINI-1.5 PRO | 48.8 | 49.6 | 28.8 | 58.6 | 49.4 | 46.0 | 48.1 | 42.0 | 68.0 |
| †GEMINI-2.0 FLASH | 45.4 | 52.4 | 30.6 | 66.7 | 31.8 | 56.0 | 46.3 | 24.5 | 55.1 |
| *Proprietary Models (API)* | | | | | | | | | |
| GEMINI-1.5 FLASH | 42.1 | 49.8 | 30.8 | 53.5 | 54.4 | 37.7 | 41.0 | 31.5 | 37.8 |
| GEMINI-1.5 PRO | 45.4 | 56.2 | 30.9 | 64.1 | 43.6 | 51.3 | 46.3 | 36.0 | 34.6 |
| GPT-4O | 34.0 | 46.2 | 5.3 | 43.8 | 38.2 | 37.0 | 41.3 | 31.5 | 28.5 |
| *Open-source Models* | | | | | | | | | |
| LLAVA-ONEVISION-0.5B | 27.1 | 33.3 | 29.2 | 13.0 | 29.1 | 29.2 | 40.2 | 36.1 | 6.6 |
| **+SEE&TREK** | $28.7_{+1.6\%}$ | $49.0_{+16.3\%}$ | $29.4_{+0.2\%}$ | $15.1_{+2.1\%}$ | $27.7_{-1.4\%}$ | $30.3_{+1.1\%}$ | $37.3_{-2.9\%}$ | $35.137.3_{-1.0\%}$ | $6.1_{-0.5\%}$ |
| LLAVA-ONEVISION-7B | 31.4 | 34.7 | 20.6 | 47.3 | 18.1 | 40.4 | 32.4 | 32.5 | 24.9 |
| **+SEE&TREK** | $33.0_{+1.6\%}$ | $32.0_{-2.7\%}$ | $17.0_{-3.6\%}$ | $39.8_{-7.5\%}$ | $27.8_{+9.7\%}$ | $39.0_{-1.4\%}$ | $40.6_{+8.2\%}$ | $31.9_{-0.6\%}$ | $35.7_{+10.8\%}$ |
| LLAVA-NEXT-VIDEO-7B | 32.5 | 37.0 | 12.1 | 45.6 | 26.4 | 35.1 | 39.3 | 34.0 | 31.1 |
| **+SEE&TREK** | $33.8_{+1.3\%}$ | $39.2_{+2.2\%}$ | $11.9_{-0.2\%}$ | $47.5_{+1.9\%}$ | $26.6_{+0.2\%}$ | $39.9_{+4.8\%}$ | $40.9_{+1.6\%}$ | $36.6_{+2.6\%}$ | $28.3_{-2.8\%}$ |
| INTERNVL3-1B | 29.5 | 65.0 | 18.5 | 15.9 | 22.5 | 29.3 | 47.8 | 27.3 | 9.8 |
| **+SEE&TREK** | $32.0_{+3.5\%}$ | $63.6_{-1.4\%}$ | $25.8_{+7.4\%}$ | $16.1_{+0.2\%}$ | $30.7_{+8.2\%}$ | $32.8_{+10.0\%}$ | $46.4_{-1.4\%}$ | $28.4_{+1.1\%}$ | $12.3_{+2.5\%}$ |
| INTERNVL3-8B | 40.2 | 67.8 | 32.2 | 44.5 | 41.8 | 42.8 | 37.7 | 26.3 | 28.3 |
| **+SEE&TREK** | $43.2_{+3.0\%}$ | $65.2_{-2.6\%}$ | $32.9_{+0.7\%}$ | $46.9_{+2.4\%}$ | $46.7_{+4.9\%}$ | $45.9_{+3.1\%}$ | $40.2_{+2.5\%}$ | $30.4_{+4.1\%}$ | $37.4_{+9.1\%}$ |
| INTERNVL3-14B | 44.2 | 69.0 | 33.6 | 53.7 | 45.6 | 43.9 | 42.4 | 23.7 | 41.4 |
| **+SEE&TREK** | $45.6_{+1.4\%}$ | $65.9_{-3.1\%}$ | $35.7_{+2.1\%}$ | $50.5_{-2.2\%}$ | $48.4_{+2.8\%}$ | $49.0_{+5.1\%}$ | $41.0_{-1.4\%}$ | $27.8_{+4.1\%}$ | $46.8_{+5.4\%}$ |
| QWEN2.5-VL-3B | 25.7 | 15.0 | 17.4 | 16.0 | 27.0 | 35.1 | 44.6 | 29.9 | 21.1 |
| **+SEE&TREK** | $26.7_{+1.0\%}$ | $9.7_{-5.3\%}$ | $23.7_{+6.3\%}$ | $19.0_{+3.0\%}$ | $22.7_{-4.3\%}$ | $33.2_{-1.9\%}$ | $47.0_{+2.4\%}$ | $29.4_{-0.5\%}$ | $28.8_{+7.7\%}$ |
| QWEN2.5-VL-7B | 27.3 | 13.0 | 14.4 | 35.9 | 21.3 | 36.9 | 37.9 | 29.9 | 29.6 |
| **+SEE&TREK** | $29.0_{+2.6\%}$ | $13.6_{+0.6\%}$ | $14.7_{+0.3\%}$ | $35.4_{-0.5\%}$ | $23.6_{+2.3\%}$ | $33.4_{-3.5\%}$ | $41.3_{+3.4\%}$ | $30.4_{+0.5\%}$ | $39.2_{+9.6\%}$ |
| QWEN2.5-VL-32B | 33.7 | 16.7 | 22.6 | 47.0 | 37.8 | 39.3 | 37.6 | 29.9 | 38.7 |
| **+SEE&TREK** | $34.7_{+1.0\%}$ | $19.9_{+3.2\%}$ | $23.8_{+1.2\%}$ | $41.0_{-6.0\%}$ | $39.5_{+1.7\%}$ | $36.9_{-2.4\%}$ | $39.2_{+1.6\%}$ | $32.5_{+2.6\%}$ | $44.9_{+6.2\%}$ |
| *KIMI-VL-A3B | 33.4 | 24.2 | 30.8 | 49.6 | 33.5 | 33.8 | 39.7 | 28.9 | 27.2 |
| **+SEE&TREK** | $35.1_{+1.7\%}$ | $23.5_{-0.7\%}$ | $30.4_{-0.4\%}$ | $48.4_{-1.2\%}$ | $38.3_{+4.8\%}$ | $35.6_{+1.8\%}$ | $41.9_{+2.2\%}$ | $30.9_{+2.0\%}$ | $31.9_{+4.7\%}$ |

## 3.3 Joint Optimized Prompting

We leverage the enhanced keyframes obtained from Maximum Semantic Richness Sampling (Sec. 3.1) alongside RGB visualizations of the BEV and 3D trajectories generated from Motion Reconstruction (Sec. 3.2) to construct a unified visual input. These elements are then combined with a carefully designed textual prompt that includes: 1) general descriptions of the selected keyframes and their semantic properties, 2) references to the BEV and 3D trajectory visualizations, and (3) the relative coordinates of key points along the camera path. Together with a target spatial reasoning question, these components are integrated into an instruction-style prompt. This heuristic formulation serves to inject explicit spatial cues into the MLLM's input, enhancing its ability to reason about geometric layout, motion progression, and scene structure in a lightweight, model-agnostic manner. Note that our method is a training- and GPU-free method, which just needs to compute several times on the CPU and requires only a single forward pass.

# 4 Experiments

*Implementation and more comprehensive experiments details can be found in Appendix B.*

## 4.1 Evaluation Setup

**Datasets.** We select VSI-BENCH [38] and STI-BENCH [39] as our spatial evaluation benchmark. **1) VSI-BENCH** is a very challenging benchmark that requires understanding spatial relationships and correspondences of multiple objects in a video [38]. It comprises over 5,000 question-answer pairs derived from 288 real videos, with duration ranging from 1.5 minutes to 2.5 minutes. These videos are sourced from the validation sets of the public indoor 3D scene reconstruction datasets ScanNet [45], ScanNet++ [46], and ARKitScenes [47] and represent diverse environments—including residential spaces, professional settings (*e.g.*, offices, labs), and industrial spaces (e.g., factories)—and multiple geographic regions. Compared to normal multimodal spatial benchmarks like SCANQA [14], VSI-BENCH has a wider range of random changes in viewing angles. **2) STI-BENCH** is a benchmark designed to evaluate MLLMs' spatial-temporal understanding through challenging tasks such as estimating and predicting the appearance, pose, displacement, and motion of objects. It contains 2,064 QA pairs across desktop, indoor, and outdoor scenarios, providing a systematic quantitative assessment of MLLMs' spatial-temporal understanding capabilities.

Table 2: Comparison of various MLLMs boosting by **SEE&TREK** on the STI-BENCH.

| Methods | Avg. | Dim. Meas. | Spatial Rel. | 3D Video Grounding | Disp. & P.L. | Speed & Acc. | Ego & Orient. | Traj. Desc. | Pose Est. |
|---|---|---|---|---|---|---|---|---|---|
| | | Static Understanding | | | Dynamic Understanding | | | | |
| *Proprietary Models (API)* | | | | | | | | | |
| GPT-4o | 34.8 | 24.9 | 49.6 | 28.1 | 27.6 | 36.0 | 30.3 | 36.8 | 51.3 |
| Claude-3.7-Sonnet | 39.4 | 31.8 | 49.0 | 36.3 | 29.0 | 36.9 | 27.0 | 41.0 | 62.7 |
| Gemini-2.0-Flash | 38.7 | 33.7 | 50.0 | 33.7 | 32.7 | 34.4 | 15.1 | 48.7 | 62.4 |
| Gemini-2.5-Pro | 40.9 | 34.2 | 53.4 | 32.3 | 32.4 | 34.3 | 44.9 | 52.0 | 58.4 |
| *Open-source Models* | | | | | | | | | |
| INTERNVL3-1B | 18.7 | 19.4 | 19.2 | 18.6 | 19.3 | 18.8 | 8.1 | 24.4 | 21.7 |
| +SEE&TREK | $20.4_{+1.7\%}$ | $25.0_{+5.6\%}$ | $17.4_{-1.8\%}$ | $20.5_{+1.9\%}$ | $18.2_{-1.1\%}$ | $19.7_{+0.9\%}$ | $12.6_{+4.5\%}$ | $22.4_{-2.0\%}$ | $24.4_{+2.7\%}$ |
| INTERNVL3-8B | 30.2 | 27.0 | 36.3 | 28.7 | 25.2 | 32.7 | 23.8 | 25.6 | 42.2 |
| +SEE&TREK | $31.2_{+1.0\%}$ | $26.7_{-0.3\%}$ | $36.4_{+0.1\%}$ | $34.1_{+5.4\%}$ | $21.0_{-3.8\%}$ | $35.4_{+2.7\%}$ | $27.9_{+4.1\%}$ | $26.3_{+0.7\%}$ | $42.7_{+0.5\%}$ |
| INTERNVL3-14B | 30.8 | 27.7 | 41.1 | 32.2 | 19.1 | 28.8 | 17.3 | 25.6 | 49.4 |
| +SEE&TREK | $32.2_{+1.4\%}$ | $29.9_{+2.2\%}$ | $40.3_{-0.8\%}$ | $32.2_{+0.0\%}$ | $22.4_{+3.3\%}$ | $30.5_{+1.7\%}$ | $19.7_{+2.4\%}$ | $34.2_{+8.6\%}$ | $48.0_{-1.4\%}$ |
| QWEN2.5-VL-7B | 35.6 | 25.3 | 52.1 | 33.4 | 19.9 | 31.5 | 41.6 | 50.0 | 52.2 |
| +SEE&TREK | $36.9_{+1.3\%}$ | $24.2_{-1.1\%}$ | $45.2_{-6.9\%}$ | $35.3_{+1.9\%}$ | $20.5_{+0.6\%}$ | $32.1_{+0.6\%}$ | $57.8_{+16.2\%}$ | $48.7_{-1.3\%}$ | $52.2_{+0.0\%}$ |
| QWEN2.5-VL-32B | 40.5 | 36.3 | 46.6 | 39.7 | 33.6 | 40.0 | 22.7 | 44.8 | 57.2 |
| +SEE&TREK | $41.7_{+1.2\%}$ | $36.7_{+0.4\%}$ | $49.3_{+2.7\%}$ | $37.2_{-2.5\%}$ | $31.7_{-1.9\%}$ | $42.4_{+2.4\%}$ | $33.5_{+10.8\%}$ | $46.2_{+1.4\%}$ | $58.9_{+1.7\%}$ |

**Benchmark Models.** Following [38], we comprehensively evaluate 10 video-supporting open-source MLLMs across diverse model families on our proposed **SEE&TREK**, encompassing various parameter scales, training recipes, and model architectures. For proprietary models, we consider Gemini-1.5 [4] and GPT-4o [3] for comparison. For open-source models, we evaluate models from InternVL3 [48], LLaVA-OneVision [49], LLaVA-NeXT-Video [6], Qwen2.5 [1] and Kimi-VL [37]. All evaluations are conducted under zero-shot settings. To ensure reproducibility, we use greedy decoding for all models. In VSI-BENCH evaluation, *Baseline* and *VSI-Bench (tiny) Perf.* are borrowed from [38], which are only utilized for comparison.

## 4.2 Main Results

**1) VSI-BENCH.** As shown in Table 1, **SEE&TREK** consistently enhances the performance of various open-source multimodal models on the VSI-BENCH benchmark. In terms of overall accuracy (Avg.), all tested models benefit from the integration of **SEE&TREK**, with gains ranging from +1.0% to +3.5%. The most notable improvement is observed on INTERNVL3-1B, which achieves a +3.5% boost, highlighting the effectiveness of **SEE&TREK** for relatively smaller models. Regarding specific task types, **SEE&TREK** notably improves numerical reasoning (*e.g.*, Abs. Dist.) and spatial understanding (*e.g.*, Route Plan and Appr. Order), where we observe significant relative gains such as +10.8% in Appr. Order for LLAVA-ONEVISION-7B. These trends suggest that **SEE&TREK** is particularly beneficial for complex spatial-temporal reasoning tasks. We also observe that larger models (*e.g.*, INTERNVL3-14B, QWEN2.5-VL-32B) tend to show modest but consistent improvements. This indicates that while **SEE&TREK** helps models of all sizes, its relative impact is more pronounced on lightweight or mid-sized MLLMs, potentially due to their higher reliance on external structural cues to compensate for limited capacity. It is also worth noting that some performance drops (highlighted in green) appear in isolated tasks (*e.g.*, Rel. Dir. and Obj. Count) for certain models, which might be attributed to trade-offs introduced by the augmented perception pipeline or model-specific biases. However, the overall trend strongly favors the inclusion of **SEE&TREK** as a plug-and-play enhancement module for open-source MLLMs in video-centric spatial reasoning tasks.

**2) STI-BENCH.** As shown in Table 2, the integration of **SEE&TREK** consistently enhances overall accuracy across all evaluated open-source models on the STI-BENCH. For the INTERNVL3 series, they achieve respective improvements of +1.7%, +1.0%, and +1.4% in average performance respectively. This consistent gain across models of varying scales underscores the scalability and robustness of **SEE&TREK**. At the sub-task level, **SEE&TREK** yields particularly notable improvements in dynamic understanding. For instance, INTERNVL3-14B shows substantial gains in Trajectory Description (+8.6%) and Displacement & Path Length (+3.3%), reflecting enhanced temporal and spatial tracking capabilities. Likewise, INTERNVL3-8B benefits significantly in 3D Video Grounding (+5.4%) and Ego & Orientation (+2.7%). These results demonstrate that **SEE&TREK** is especially effective at reinforcing temporal-spatial reasoning. Although minor performance declines are observed in a few categories (*e.g.*, Room Size or Relative Direction), they are marginal and do not offset the substantial improvements in key dynamic tasks.

Table 3: Ablation studies of Maximum Semantic Richness Sampling and Motion Reconstruction.

| Max. Semantic Rich. Sampling | Motion Reconstruction | Avg. | Obj. Count | Abs. Dist. | Obj. Size | Room Size | Rel. Dist. | Rel. Dir. | Route Plan | Appr. Order |
|---|---|---|---|---|---|---|---|---|---|---|
| | | | | Numerical Answer | | | | Multiple-Choice Answer | | |
| | | 40.2 | 67.8 | 32.2 | 44.5 | 41.8 | 42.8 | 37.7 | 26.3 | 28.3 |
| ✓ | | 41.8 | 67.3 | 34.7 | 39.0 | 47.5 | 43.9 | 44.3 | 25.8 | 31.9 |
| | ✓ | 42.1 | 64.6 | 29.7 | 46.4 | 44.7 | 39.1 | 40.7 | 28.8 | 43.0 |
| ✓ | ✓ | 43.2 | 65.2 | 32.9 | 46.9 | 46.7 | 45.9 | 40.2 | 30.4 | 37.4 |

Table 4: Ablation studies of SPATIOTEMPORAL ENCODING and point prompts.

| SPATIOTEMPORAL ENCODING | point prompts | Avg. | Obj. Count | Abs. Dist. | Obj. Size | Room Size | Rel. Dist. | Rel. Dir. | Route Plan | Appr. Order |
|---|---|---|---|---|---|---|---|---|---|---|
| | | | | Numerical Answer | | | | Multiple-Choice Answer | | |
| | | 42.4 | 64.9 | 32.8 | 45.3 | 46.9 | 44.9 | 40.5 | 27.3 | 37.1 |
| ✓ | | 42.7 | 66.3 | 34.7 | 43.5 | 46.9 | 46.9 | 39.1 | 31.4 | 34.7 |
| | ✓ | 42.7 | 66.5 | 35.0 | 44.1 | 44.9 | 46.6 | 38.9 | 29.9 | 35.4 |
| ✓ | ✓ | 43.2 | 65.2 | 32.9 | 46.9 | 46.7 | 45.9 | 40.2 | 30.4 | 37.4 |

## 4.3 Ablation Studies

In this section, we utilize VSI-BENCH for evaluation and leverage INTERNVL3-8B as the baseline model. We investigate the effect of each technique proposed in our **SEE&TREK**. *More investigation can be found in the Appendix.*

**Overall.** To systematically assess the contribution of different spatial cues, we first conduct ablation studies on two components: Maximum Semantic Richness Sampling (dubbed MSRS) and Motion Reconstruction. As shown in Table 3, enabling MSRS alone leads to a moderate performance gain (average accuracy from 40.2% to 41.8%), with notable improvements observed in static layout-related tasks such as object size. This suggests that semantically richer keyframe selection contributes to more informative spatial representations. Besides, incorporating Motion Reconstruction yields distinct advantages in dynamic reasoning tasks, particularly relative direction and route planning (*e.g.*, route planning improves from 26.3% to 28.8%, highlighting its role in modeling egocentric movement and temporal coherence. When both components are jointly applied, the model achieves the highest overall accuracy (43.2%), demonstrating their complementary effects in facilitating both static and dynamic aspects of spatial understanding within MLLMs.

**Optimized Prompting.** Then, we investigate the impact of SPATIOTEMPORAL ENCODING and point prompts added in instruction from Section 3.3. As shown in Table 4, adding SPATIOTEMPORAL ENCODING improves the average accuracy to 42.7%, with notable gains in tasks like Object Count (66.3%) and Relative Direction (46.9%), highlighting its role in enhancing spatial and temporal reasoning which connecting the MSRS and Motion Reconstruction. Similarly, incorporating point prompts also achieves an average accuracy of 42.7% demonstrating its effectiveness in providing explicit spatial cues. When both modules are combined, the model achieves the highest average accuracy of 43.2%, with significant improvements in diverse tasks.

**Sample Efficiency Analysis.** We conduct experiments with different sample intervals $N \in (1, 2, 3, 4, 8, 12)$ from a single forward process to analyze their impact on the efficiency performance of **SEE&TREK**. While the finest granularity ($N = 1$) achieves strong results (Avg. 42.9%) at the cost of high computational time (410s), increasing $N$ substantially reduces runtime while maintaining competitive accuracy. $N = 3$ and $N = 4$ strike the best balance, achieving the highest average score (43.2%) with over 65% reduction in processing time compared to $N = 1$. Even with larger intervals like $N = 8$ or $N = 12$, the model retains robust spatial understanding, with only a marginal drop in performance. These results demonstrate that **SEE&TREK** is highly sample-efficient, maintaining strong accuracy with significantly fewer frames and minimal computational overhead. *Note that the time consuming of* **SEE&TREK** *still largely depends the length of the given videos. Particularly, the duration of videos in* VSI-BENCH *at least longer than 1 minute.*

**Rank Selection.** We explore different rank extraction methods' impact on MSRS. Here, we utilize other two kinds of Top$K$ selection methods: 1) Original *TopK* only extracts the frames according to most object numbers. 2) *Temporal-TopK* temporally divides a video into multiple consecutive frame groups based on the number of keyframes and then performs *TopK* selection within each group. As shown in Table 6, *Balanced-TopK* consistently outperforms both *TopK* and *Temporal-TopK* across various spatial reasoning tasks, demonstrating its advantage in capturing both comprehensive and diverse visual semantics. Notably, it achieves the highest average accuracy (43.2%) and excels in multiple-choice tasks such as Relative Distance (45.9%) and Approach Order (37.4%), indicating

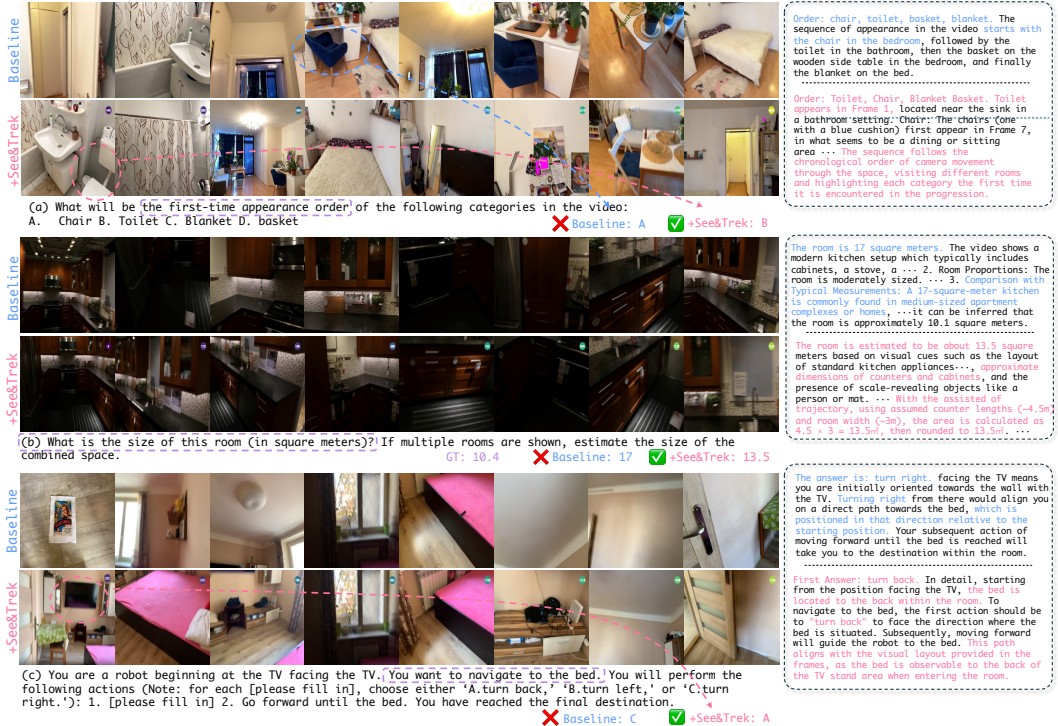

Figure 3: Qualitative results about **SEE&TREK** evaluated on VSI-BENCH. Here, we represent the different tasks of (a) appearance order (b) room size, and (c) route plan. *More results can be found in the appendix.* It shows that **SEE&TREK** obtains the visual diversity and motion reconstruction of the given video, gaining a better spatial understanding.

Table 5: Ablation studies of different sample interval $N$ settings. "Time(s)" denotes the average time consuming on processing videos from VSI-BENCH.

| $N$ | Time(s) | Avg. | Obj. Count | Abs. Dist. | Obj. Size | Room Size | Rel. Dist. | Rel. Dir. | Route Plan | Appr. Order |
|---|---|---|---|---|---|---|---|---|---|---|
| | | | **Numerical Answer** | | | | **Multiple-Choice Answer** | | | |
| 1 | 410 | 42.9 | 66.2 | 31.7 | 48.2 | 47.3 | 42.8 | 42.0 | 30.4 | 33.8 |
| 2 | 227 | 41.7 | 66.0 | 30.5 | 47.1 | 46.3 | 40.0 | 40.8 | 27.8 | 35.3 |
| 3 | 159 | 43.2 | 66.9 | 31.3 | 47.3 | 48.4 | 43.8 | 40.6 | 29.4 | 38.3 |
| 4 | 82 | 43.2 | 65.2 | 32.9 | 46.9 | 46.7 | 45.9 | 40.3 | 30.4 | 37.4 |
| 8 | 63 | 42.8 | 66.5 | 33.6 | 46.0 | 46.7 | 41.7 | 41.5 | 30.9 | 35.9 |
| 12 | 53 | 42.5 | 66.4 | 32.6 | 45.9 | 41.7 | 46.5 | 39.9 | 28.9 | 38.8 |

a better spatial understanding through a more balanced frame selection strategy. Compared with the *TopK* and *Temporal-TopK*, our proposed *Balanced-TopK* further enhances selection by jointly considering object richness, temporal distribution, and semantic diversity.

## 4.4 Qualitative results

As shown in Figure 3, we leverage self-explanations to investigate why **SEE&TREK** achieves superior performance. Thanks to the MSRS, **SEE&TREK** is able to identify the most informative frame that best represents the entire scene—such as in cases (a) and (c)—typically from the first frame, while the baseline always fails in perception and positioning. This leads to more accurate analysis in dynamic spatial understanding tasks for MLLMs. Furthermore, we observe that motion reconstruction enhances the MLLM's ability to estimate spatial distances with lower error, as demonstrated in case (b), like "with the assisted of trajectory," thereby supporting static spatial reasoning. Overall, **SEE&TREK** significantly enhances the comprehensive spatial understanding capabilities of MLLMs.

Table 6: Ablation studies of different rank extraction methods in MSRS.

| Rank Extraction | Avg. | Obj. Count | Abs. Dist. | Obj. Size | Room Size | Rel. Dist. | Rel. Dir. | Route Plan | Appr. Order |
|---|---|---|---|---|---|---|---|---|---|
| | | Numerical Answer | | | | Multiple-Choice Answer | | | |
| *TopK* | 37.9 | 65.9 | 28.7 | 42.6 | 35.5 | 37.1 | 39.2 | 28.9 | 25.7 |
| *Temporal-TopK* | 42.7 | 69.0 | 31.5 | 47.0 | 45.5 | 41.0 | 39.7 | 31.4 | 36.9 |
| *Balanced-TopK* | 43.2 | 65.2 | 32.9 | 46.9 | 46.7 | 45.9 | 40.2 | 30.4 | 37.4 |

## 5 Conclusion

We propose **SEE&TREK**, the first training- & GPU-free prompting framework designed to enhance the spatial understanding capabilities of MLLMs. We focus on enhancing the spatial understanding ability of MLLMs from two aspects: Visual Diversity and Motion Reconstruction. For achieving Visual Diversity, we conduct Maximum Semantic Richness Sampling, which employs an off-the-shell perception model, *e.g.* YOLO to efficiently extract frames that maximize semantic richness, then propose *Balanced-TopK* strategy for selecting frames. For Motion Reconstruction, we simulate visual trajectories and encode relative spatial positions to preserve both spatial relations and temporal coherence. Our method is GPU-free, requiring only a single forward pass, and can be seamlessly integrated into existing MLLMs. Comprehensive experiments on various MLLMs and two hard spatial benchmarks verify the **SEE&TREK** superiority in boosting spatial intelligence.

# A Methods

## A.1 Detailed Algorithm

The detailed algorithm of **SEE&TREK** could be found as follows:

---

**Algorithm 2** SEE&TREK: Maximum Semantic Richness Sampling and Motion Reconstruction

---

**Require:** Video $V = \{f_t\}_{t=1}^{N_v}$, sampling interval $N$, number of keyframes $K$, object detector $\mathcal{Y}(\cdot)$, camera intrinsics $\mathbf{K}$
**Ensure:** Selected keyframes $\{\mathcal{F}'_{t_i}\}_{i=1}^K$, BEV trajectory $\mathbf{P}_{\text{BEV}}$, 3D trajectory $\mathbf{P}_{\text{3D}}$, 3D trajectory set $\mathcal{T}^{\text{world}}$
1: Sample frames: $\mathcal{S} \leftarrow \{f_t \mid t \bmod N = 0\}$
2: Initialize: $\mathcal{O} \leftarrow \emptyset$, Trajectory $\mathcal{T}^{\text{world}} \leftarrow []$
3: **for** each consecutive pair $(f_{t-1}, f_t) \in \mathcal{S}$ **do**
4:      Convert $(f_{t-1}, f_t)$ to grayscale
5:      Extract ORB keypoints/descriptors: $\text{ORB}(f_{t-1}), \text{ORB}(f_t)$
6:      Match descriptors to obtain $\mathcal{M}_{t-1,t}$
7:      Normalize matched points: $\hat{\mathbf{x}}_i \leftarrow \mathbf{K}^{-1}[\mathbf{x}_i^\top\ 1]^\top$
8:      Estimate essential matrix $\mathbf{E}$ via RANSAC
9:      Decompose $\mathbf{E} \rightarrow (\mathbf{R}_t, \mathbf{T}_t)$
10:     Update global pose: $\mathbf{R}_t^{\text{world}}, \mathbf{T}_t^{\text{world}}$
11:     Append $\mathbf{T}_t^{\text{world}}$ to $\mathcal{T}^{\text{world}}$
12:     Detect objects: $\mathcal{C}_t \leftarrow \mathcal{Y}(f_t)$
13:     Append $(t, \mathcal{C}_t)$ to $\mathcal{O}$
14: **end for**
15: Filter valid interval $[t_s, t_e]$: $\mathcal{O}^* \leftarrow \{(t_i, \mathcal{C}_{t_i}) \in \mathcal{O} \mid t_s \leq t_i \leq t_e\}$
16: **Balanced-Top$K$ Selection**:
17: Select global-rich frame: $t_g \leftarrow \arg\max |\mathcal{C}_{t_i}|$
18: $\mathcal{C}_{\text{sel}} \leftarrow \mathcal{C}_{t_g}, \mathcal{T}_{\text{sel}} \leftarrow \{t_g\}$
19: Divide $\mathcal{O}^*$ into $K-1$ segments $\{\mathcal{O}^{(1)}, \ldots, \mathcal{O}^{(K-1)}\}$
20: **for** $k = 1$ to $K - 1$ **do**
21:     $t_k \leftarrow \arg\min_{(t_j, \mathcal{C}_{t_j}) \in \mathcal{O}^{(k)}} \left(|\mathcal{C}_{t_j} \cap \mathcal{C}_{\text{sel}}|,\ -|\mathcal{C}_{t_j}|,\ t_j\right)$
22:     $\mathcal{C}_{\text{sel}} \leftarrow \mathcal{C}_{\text{sel}} \cup \mathcal{C}_{t_k}, \mathcal{T}_{\text{sel}} \leftarrow \mathcal{T}_{\text{sel}} \cup \{t_k\}$
23: **end for**
24: **Spatiotemporal Encoding**:
25: **for** each $t_i \in \mathcal{T}_{\text{sel}}$ **do**
26:     Compute color $\mathbf{c}_i \leftarrow \text{ColorMap}(t_i/K)$
27:     Overlay index and color on $f_{t_i} \Rightarrow \mathcal{F}'_{t_i}$
28: **end for**
29: **Trajectory Visualization:** $\mathbf{P}_{\text{BEV}} \leftarrow \text{ProjectXY}(\mathcal{T}^{\text{world}})$, $\mathbf{P}_{\text{3D}} \leftarrow \text{Render3D}(\mathcal{T}^{\text{world}})$

---

## A.2 Motion Reconstruction

In this section, we recall the mathematical context of Visual Odometry (VO). *Note that most of these can be found in the textbooks.* The first step of VO is need to conduct ORB (Oriented FAST and Rotated BRIEF) Feature Detection and Description, then Feature Matching, finally perform Essential Matrix Estimation. For this part of algorithm development, we leverage OpenCV [2] for efficient deployment.

**ORB Feature Detection and Description.** In detailed, ORB combines FAST keypoint detection with orientation-augmented BRIEF descriptors, producing a scale- and rotation-invariant feature. Regarding **Keypoint Detection (FAST)**: Given a grayscale image $f_t : \Omega \subset \mathbb{R}^2 \rightarrow \mathbb{R}$, the FAST corner detector selects a pixel $\mathbf{p} \in \Omega$ as a keypoint if there exists a contiguous arc of $n$ pixels on the Bresenham circle $C(\mathbf{p})$ of radius 3 around $\mathbf{p}$, such that:

$$|f_t(\mathbf{q}) - f_t(\mathbf{p})| > \tau, \quad \forall \mathbf{q} \in C_n(\mathbf{p}), \tag{6}$$

where $C_n(\mathbf{p}) \subset C(\mathbf{p})$ is a contiguous segment of $n$ pixels, and $\tau$ is a contrast threshold. Typically, $n = 12$ and $|C(\mathbf{p})| = 16$. This step yields the raw keypoint set $\mathcal{K}_t = \{\mathbf{p}_i^t \in \Omega\}_{i=1}^N$, where $N$ is the

---

[2]`https://github.com/opencv/opencv-python`

number of detected corners in frame $f_t$. Regarding **Orientation Assignment**, to achieve rotation invariance, ORB computes the orientation angle $\theta_i$ of each keypoint $\mathbf{p}_i^t$ by using intensity moments of a patch $P_i$ around the keypoint:

$$\theta_i = \arctan\left(\frac{m_{01}}{m_{10}}\right), \quad m_{pq} = \sum_{\mathbf{q} \in P_i} x^p y^q f_t(\mathbf{q}), \tag{7}$$

where $(x, y)$ are coordinates relative to the keypoint $\mathbf{p}_i^t$. Finally, we perform **Descriptor Computation (BRIEF)**. The BRIEF descriptor $\mathcal{D}_t \in {0, 1}^{N \times D}$ is constructed by binary intensity comparisons between $D/2$ pre-defined point pairs $(\mathbf{a}_k, \mathbf{b}_k)$ in the local patch around each $\mathbf{p}_i^t$, rotated by angle $\theta_i$:

$$\mathcal{D}_t(i, k) = \begin{cases} 1 & \text{if } f_t(R_{\theta_i}\mathbf{a}_k + \mathbf{p}_i^t) < f_t(R_{\theta_i}\mathbf{b}_k + \mathbf{p}_i^t), \\ 0 & \text{otherwise}, \end{cases} \tag{8}$$

where $R_{\theta_i} \in SO(2)$ is the 2D rotation matrix corresponding to $\theta_i$.

**Feature Matching.** Given two sets of ORB descriptors $\mathcal{D}_{t-1}, \mathcal{D}_t \in {0, 1}^{N \times D}$, feature correspondences are established by computing Hamming distances between binary descriptor vectors. For each descriptor $\mathbf{d}i^{t-1} \in \mathcal{D}t - 1$, define the matching descriptor in frame $t$ as:

$$\mathbf{d}_j^t = \arg\min_{\mathbf{d} \in \mathcal{D}_t} \text{Hamming}(\mathbf{d}_i^{t-1}, \mathbf{d}), \tag{9}$$

and accept the match if the distance is below a threshold $\tau_d$ or passes Lowe's ratio test. Let the resulting matched keypoint pairs be:

$$\mathcal{M}_{t-1,t} = \left\{(\mathbf{x}_i^{t-1}, \mathbf{x}_i^t)\right\}_{i=1}^P, \quad \mathbf{x}_i^{t-1}, \mathbf{x}_i^t \in \mathbb{R}^2, \tag{10}$$

with $\mathbf{x}i^{t-1} = \mathbf{p}a_i^{t-1}$ and $\mathbf{x}i^t = \mathbf{p}b_i^t$ corresponding to matched keypoints.

**Essential Matrix Estimation.** Given matched pixel coordinates $(\mathbf{x}_i^{t-1}, \mathbf{x}i^t)i = 1^P$ and the intrinsic matrix $\mathbf{K} \in \mathbb{R}^{3 \times 3}$, normalized coordinates are computed:

$$\hat{\mathbf{x}}_i^{t-1} = \mathbf{K}^{-1}\begin{bmatrix} \mathbf{x}_i^{t-1} \\ 1 \end{bmatrix}, \quad \hat{\mathbf{x}}_i^t = \mathbf{K}^{-1}\begin{bmatrix} \mathbf{x}_i^t \\ 1 \end{bmatrix}, \quad \hat{\mathbf{x}} \in \mathbb{R}^3. \tag{11}$$

The essential matrix $\mathbf{E} \in \mathbb{R}^{3 \times 3}$ encodes the relative motion such that $(\hat{\mathbf{x}}_i^t)^\top \mathbf{E} \hat{\mathbf{x}}_i^{t-1} = 0, \forall i$. In matrix terms, stacking all equations gives:

$$\mathbf{A} \cdot \text{vec}(\mathbf{E}) = 0, \text{ where } \mathbf{A} \in \mathbb{R}^{P \times 9}. \tag{12}$$

This is solved via SVD of $\mathbf{A}$ in the 8-point algorithm, or a robust estimator such as:

$$\mathbf{E} = \arg\min_{\mathbf{E}'} \sum_i \rho\left((\hat{\mathbf{x}}_i^t)^\top \mathbf{E}' \hat{\mathbf{x}}_i^{t-1}\right), \tag{13}$$

where $\rho(\cdot)$ is a robust loss (e.g., truncated quadratic), and outliers are rejected via RANSAC. Since $\mathbf{E}$ must satisfy the singular value constraint $\sigma_1 = \sigma_2, \sigma_3 = 0$, we project it onto the essential matrix manifold via SVD:

$$\mathbf{E} = \mathbf{U} \cdot \text{diag}(1, 1, 0) \cdot \mathbf{V}^\top. \tag{14}$$

Finally, we decompose the essential matrix to motion. Specially, The essential matrix relates to the relative pose via: $\mathbf{E} = [\mathbf{T}_t]_\times \mathbf{R}_t$, where $[\cdot]_\times$ denotes the skew-symmetric matrix:

$$[\mathbf{T}_t]_\times = \begin{bmatrix} 0 & -T_z & T_y \\ T_z & 0 & -T_x \\ -T_y & T_x & 0 \end{bmatrix}. \tag{15}$$

The decomposition of $\mathbf{E}$ via SVD provides four candidate solutions $(\pm \mathbf{R}_t, \pm \mathbf{T}_t)$, disambiguated by the Cheirality condition—checking the number of triangulated points in front of both cameras.

### A.3 Joint Optimized Prompting

Here, we give the sample of our general input containing text and vision for MLLMs evaluation. For INTERNVL3-8B, after spatial prompting from **SEE&TREK**, the visual inputs are as shown in Figure 4, then we also design the corresponding text prompt template as shown in A.3. "Points" denotes the relative spatial coordinates of each selected frames. Since different MLLMs have different training data, architectures, and training methods, using text prompts or point coordinates as prompts has a particularly large impact on performance. Therefore, we fine-tune the instructions for different model series to enable **SEE&TREK** to fully utilize its spatial understanding advantages.

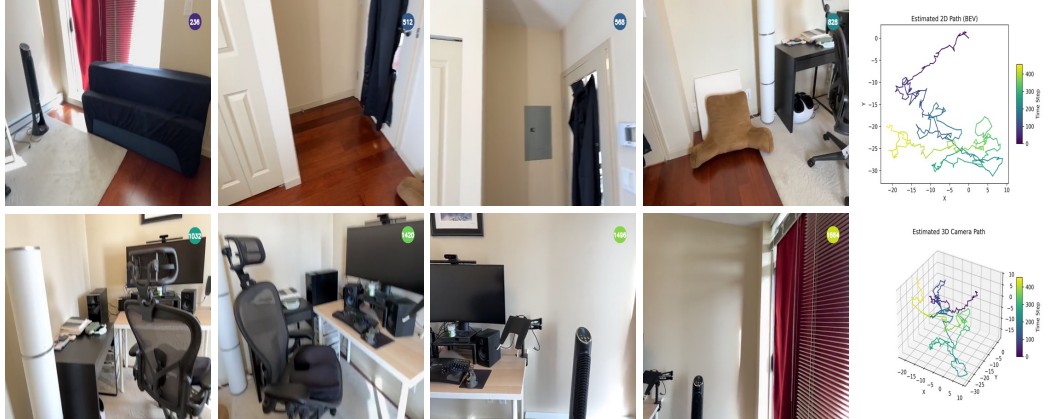

Figure 4: The sample of the visual prompting input in **SEE&TREK**.

```
1  spatial_prompt_universal = (
2      "Each video frame has its serial number in the top-right corner. "
3      "The highlight color mark of frame matches the color in the
           spatial map, indicating its position."
4  )
5  spatial_prompt_2D_3D = (
6      "Both 2D (bird's-eye) and 3D views illustrate the camera's spatial
           trajectory, "
7      "with color encoding time progression."
8  )
9  spatial_prompt_points = (
10     "Points represent the camera's relative positions; the number of
           points reflects only spatial relationships."
11 )
12 input_prompt = spatial_prompt_universal + spatial_prompt_2D_3D +
       spatial_prompt_points + points + "\n" + question
```

Figure 5: The sample of the text prompting input in **SEE&TREK**. "Points" denotes the relative spatial coordinates of each selected frames.

## B  Experiments

### B.1  VSI-Bench Dataset

**Overview.** VSI-BENCH [38] includes eight tasks of three types: *configurational*, *measurement estimation*, and *spatiotemporal*. The configurational tasks (*object count*, *relative distance*, *relative direction*, *route plan*) test a model's understanding of the configuration of space and are more intuitive for humans. Measurement estimation (of *object size*, *room size*, and *absolute distance*) is of value to any embodied agent. While predicting a measurement exactly is very difficult, for both humans and models, a better sense of distance and other measurements is intuitively correlated with better visual-spatial intelligence and underpins a wide range of tasks that require spatial awareness, like interaction with objects and navigation. Spatiotemporal tasks like appearance order test a model's memory of space as seen in the video.

**Metric Design.** Based on whether the ground-truth answer is verbal or numerical, VSI-BENCH tasks are suited to either a Multiple-Choice Answer (MCA) or Numerical Answer (NA) format. For MCA tasks, VSI-BENCH follows standard practice by using *Accuracy* ($\mathcal{ACC}$), based on exact matching (with possible fuzzy matching), as the primary metric. For NA tasks, where models predict continuous values, accuracy via exact matching fails to capture the degree of proximity between model predictions and ground-truth answers. Therefore, VSI-BENCH utilizes a new metric, *Mean Relative Accuracy* ($\mathcal{MRA}$) inspired by previous works. Specifically, for a NA question, given a model's prediction $\hat{y}$, ground truth $y$, and a confidence threshold $\theta$, relative accuracy is calculated

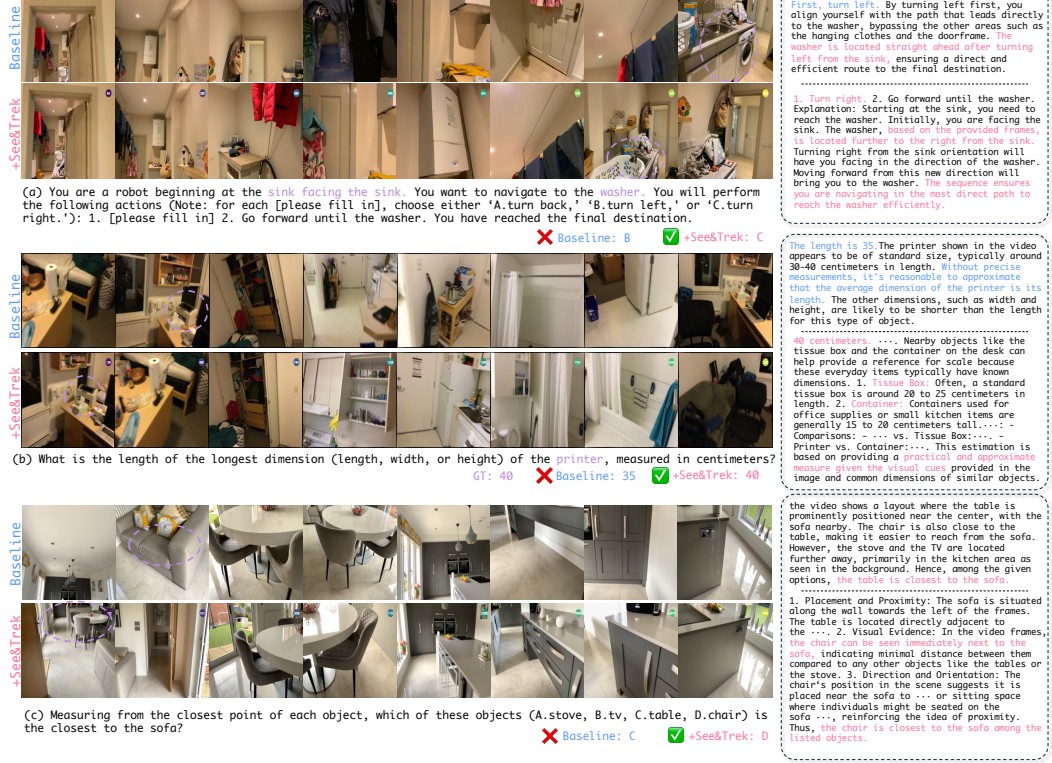

(a) You are a robot beginning at the sink facing the sink. You want to navigate to the washer. You will perform the following actions (Note: for each [please fill in], choose either 'A.turn back,' 'B.turn left,' or 'C.turn right.'): 1. [please fill in] 2. Go forward until the washer. You have reached the final destination.

❌ Baseline: B     ✅ +See&Trek: C

(b) What is the length of the longest dimension (length, width, or height) of the printer, measured in centimeters?

GT: 40     ❌ Baseline: 35     ✅ +See&Trek: 40

(c) Measuring from the closest point of each object, which of these objects (A.stove, B.tv, C.table, D.chair) is the closest to the sofa?

❌ Baseline: C     ✅ +See&Trek: D

Figure 6: Qualitative results about **SEE&TREK** evaluated on VSI-BENCH. Here, we represent the different tasks of (a) Route Plan (b) Object Absolute Size (Obj. Size), and (c) Object Relative Distance (Rel. Dist.).

by considering $\hat{y}$ correct if the relative error rate, defined as $|\hat{y} - y|/y$, is less than $1 - \theta$. As single-confidence-threshold accuracy only considers relative error in a narrow scope, $\mathcal{MRA}$ averages the relative accuracy across a range of confidence thresholds $\mathcal{C} = \{0.5, 0.55, ..., 0.95\}$:

$$\mathcal{MRA} = \frac{1}{10} \sum_{\theta \in \mathcal{C}} \mathbb{I} \left( \frac{|\hat{y} - y|}{y} < 1 - \theta \right). \tag{16}$$

$\mathcal{MRA}$ offers a more reliable and discriminative measurement for calculating the similarity between numerical predictions and ground truth values.

**Chance Level Baselines**. VSI-BENCH provides two baselines: 1) *Chance Level* (Random) is the random selection accuracy for MCA tasks (and is inapplicable for NA tasks). 2) *Chance Level* (Frequency) represents the highest performance MLLMs would achieve by always selecting the most frequent answer for each task. This identifies performance gains that may result from inherently long-tailed answers or imbalanced multiple-choice distributions.

**Human Level Performance.** VSI-BENCH randomly sample a subset of 400 questions (50 per task), which we will refer to as VSI-BENCH (tiny). Human evaluators independently answer each question, and their performance is evaluated using the above-mentioned metrics.

### B.2   STI-Bench Dataset

**Overview.** STI-BENCH [39] contains 300 videos and more than 2,000 QA pairs, covering three major scenarios: Desktop, Indoor, and Outdoor. The videos are sourced from OMNI6DPOSE [50], SCANNET [45] and WAYMO [51] respectively, thus encompassing a broad spectrum of real-world environments. They propose eight tasks in total, each one systematically examining a distinct aspect of MLLMs' spatial-temporal understanding, which divide these tasks into two main categories: 1) **Static Understanding**: *Dimensional Measurement, Spatial Relation, 3D Video Grounding*; 2)

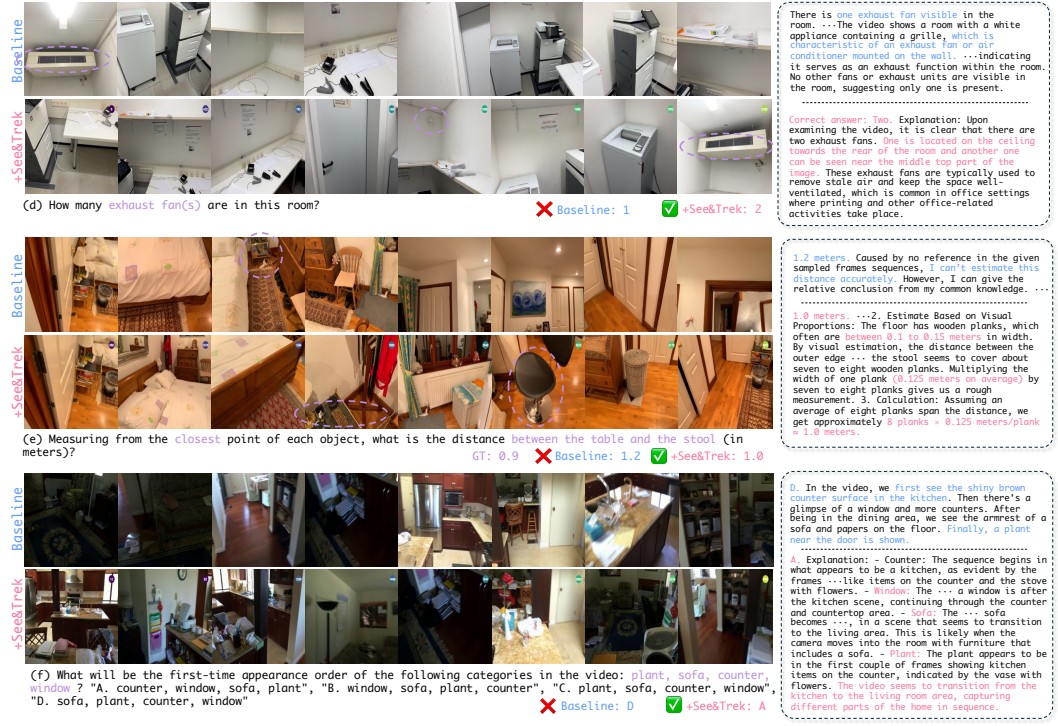

Figure 7: Qualitative results about **SEE&TREK** evaluated on VSI-BENCH. Here, we represent the different tasks of (d) Object Count (Obj. Count) (e) Object Absolute Distance (Abs. Dist.), and (f) Appearance Order (Appr. Order).

**Dynamic Understanding**: *Displacement and Path Length*, *Speed and Acceleration*, *Ego-Centric Orientation*, *Trajectory Description* and *Pose Estimation*.

**API Engine and Baseline Settings.** Regarding the commercial engine, STI-BENCH uniformly sample 30 frames from the video for each record and explicitly indicate the sampling FPS (Frames Per Second) for the current video within the prompt. An exception is made for Claude3.7-Sonnet, for which only 20 frames are sampled due to its API constraints. STI-BENCH are presented in a multiple-choice format with five possible answers, hence a random guess baseline yields a 20% accuracy.

**Metric Design.** 1) *Dimensional Measurement.* let $l_x, l_y, l_z$ denote the dimensions (length, width, height) of an object along the $x, y$, and $z$ axes:

$$l_x = x_{\max} - x_{\min}, l_y = y_{\max} - y_{\min}, l_z = z_{\max} - z_{\min}, \tag{17}$$

Here, $l_x, l_y, l_z$ represent the object size along each coordinate axis. If it needs the distance between two objects (or between the camera and an object), let $d_{12}$ be the Euclidean distance between their center points:

$$d_{12} = \sqrt{(x_2 - x_1)^2 + (y_2 - y_1)^2 + (z_2 - z_1)^2}. \tag{18}$$

Here, $(x_1, y_1, z_1)$ and $(x_2, y_2, z_2)$ are the center coordinates of the two objects.

2) *Spatial Relation.* When the difference along one coordinate axis is significantly larger than along others, the sign of that difference determines the spatial relation:

$$r_{xy} = \mathrm{sign}(x_A - x_B), r_{yz} = \mathrm{sign}(y_A - y_B), r_{zx} = \mathrm{sign}(z_A - z_B). \tag{19}$$

Here, $r_{xy}, r_{yz}, r_{zx}$ indicate relative positioning along each axis (e.g., front/back, left/right, above/below). It choose the axis with the greatest difference to label the dominant relation.

3) *3D Video Grounding.* For frame $t$ in the camera coordinate system, the 3D bounding box of an object can be described with dimensions, center position, and optional rotations:

$$\mathrm{BBox}_t = (l_t, w_t, h_t, x_t, y_t, z_t, \mathrm{yaw}_t, \mathrm{pitch}_t, \mathrm{roll}_t). \tag{20}$$

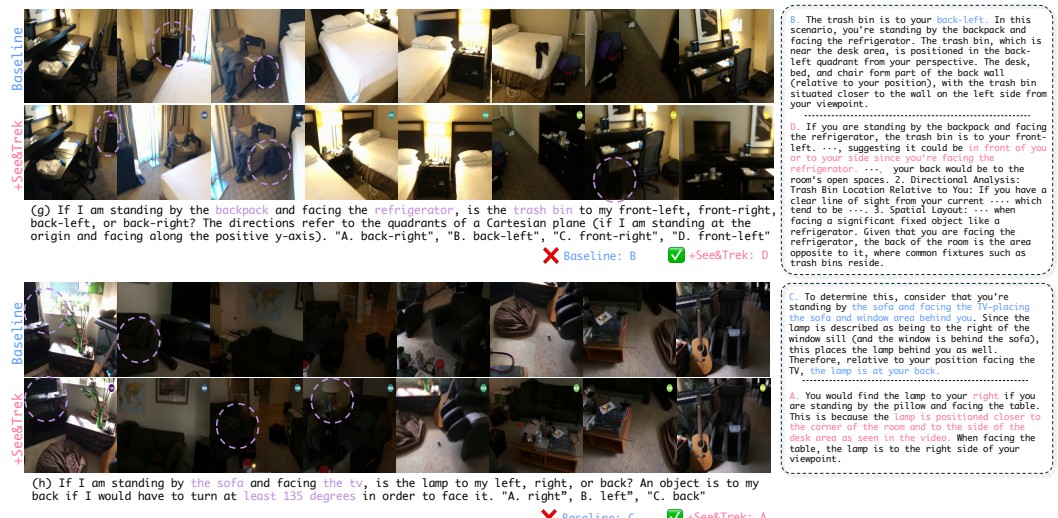

Figure 8: Qualitative results about **SEE&TREK** evaluated on VSI-BENCH. Here, we represent the different tasks of (g) Route Plan (h) Object Relative Direction (Rel. Dir.)

Here, $(l_t, w_t, h_t)$ are the object dimensions, $(x_t, y_t, z_t)$ is the center position, and $(\text{yaw}_t, \text{pitch}_t, \text{roll}_t)$ are optional rotation angles if available.

4) *Pose Estimation.* Given the camera's initial pose $(p_0, o_0)$, the pose $(p_t, o_t)$ at time $t$ can be obtained using the extrinsic-derived matrices $R_t$ (rotation) and $T_t$ (translation):

$$p_t = R_t p_0 + T_t, o_t = o_0 + \Delta o_t. \tag{21}$$

Here, $p_t$ is the position, $o_t$ is the orientation.

5) *Displacement and Path Length.* Let $p_i = (x_i, y_i, z_i)$ be the position at time $i$. The displacement $d_{0n}$ and path length $L_{\text{traj}}$ are computed as:

$$d_{0n} = \sqrt{(x_n - x_0)^2 + (y_n - y_0)^2 + (z_n - z_0)^2}, \tag{22}$$

$$L_{\text{traj}} = \sum_{i=1}^{n} \sqrt{(x_i - x_{i-1})^2 + (y_i - y_{i-1})^2 + (z_i - z_{i-1})^2}. \tag{23}$$

Here, $d_{0n}$ is the straight-line distance from the initial to the final position; $L_{\text{traj}}$ sums consecutive segment lengths for the entire path.

6) *Speed and Acceleration.* Let $\Delta t$ be the time interval between consecutive frames. Then the speed $v_i$ and acceleration $a_i$ are:

$$v_i = \frac{d_i}{\Delta t}, a_i = \frac{v_i - v_{i-1}}{\Delta t}. \tag{24}$$

Here, $d_i$ is the displacement between adjacent frames, $v_i$ is the speed at time $i$, and $a_i$ is the acceleration.

7) *Ego-Centric Orientation.* If $\theta_t$ denotes the camera orientation (azimuth) at time $t$, then the orientation change $\Delta\theta_t$ is:

$$\Delta\theta_t = \theta_t - \theta_0. \tag{25}$$

This indicates how much the camera has rotated relative to its initial azimuth.

8) *Trajectory Description.* It applies the Ramer-DouglasPeucker (RDP) algorithm to simplify the sequence of positions into key line segments. The resulting polyline is described in a piecewise manner (*e.g.*, "go straight for 30m, turn left 85°, then go straight for 20m, ..."), providing a concise representation of complex motion trajectories.

## B.3 Implementation

**SEE&TREK** focus on the pre-processing stage of MLLMs, which samples one frame for every four frames from the given spatial video. For fair evaluation, we adopt 8 frames as input to test each MLLMs for the given videos. For obtaining visual diversity and balancing efficiency, we utilize YOLOv8-Tiny, named YOLOv8N from Ultralytics[3] for faster detection. All experiments are conducted on NVIDIA 8×A6000 and 6×A800. In actual development, to accelerate the overall testing process, we first utilize Motion construction and YOLO to process each sampled frame from videos for subsequent evaluation calls, which stores the corresponding spatial information. In evaluation, we call the stored spatial information from the last step and perform the *Balanced-TopK* sampling and SPATIOTEMPORAL ENCODING techniques in each question and answer process. Note that we do not utilize the intrinsic parameters provided by the original datasets (e.g., VSI-BENCH or STI-BENCH). Instead, we just adopt a fixed intrinsic matrix derived from the KITTI dataset ($\mathbf{K} = [[718.8560, 0, 607.1928], [0, 718.8560, 185.2157], [0, 0, 1]]$) across all experiments.

## B.4 More Comparison Results

We conduct the experiments on OPENEQA [52] with InternVL3-8B to show our method's superiority. Note that we utilize the (EM-EQA) setting [52] to evaluate the performance, and we still set 8 frames as input.

Table 7: Comparison results on the OPENEQA [52].

| Model | w/o SEE&TREK | w. SEE&TREK | Gain(%) |
|---|---|---|---|
| Qwen2.5-VL-7B | 47.1 | 49.1 | +2.0% |
| InternVL3-8B | 50.5 | 52.3 | +1.8% |

## B.5 Ablation Study

In this section, we utilize VSI-BENCH for evaluation and leverage INTERNVL3-8B as the baseline model. We investigate the effect of each technique proposed in our **SEE&TREK**.

Table 8: Ablation studies of each component of *Balanced-TopK*.

| *Balanced-TopK* | Avg. | Obj. Count | Abs. Dist. | Obj. Size | Room Size | Rel. Dist. | Rel. Dir. | Route Plan | Appr. Order |
|---|---|---|---|---|---|---|---|---|---|
| | | Numerical Answer | | | | Multiple-Choice Answer | | | |
| w/o $\mathcal{C}_{sel}$ Min. | 42.5 | 69.4 | 33.5 | 45.6 | 44.6 | 41.4 | 39.9 | 28.4 | 37.4 |
| w/o Count Max. | 43.1 | 67.1 | 31.9 | 45.9 | 47.5 | 42.5 | 39.5 | 29.4 | 40.9 |
| w/o Time Early | 43.2 | 66.2 | 34.0 | 45.5 | 48.0 | 42.1 | 41.0 | 31.4 | 37.3 |

**Maximum Semantic Richness Sampling.** As mentioned in Section 3.1, we want to select one frame $\tau_k$ such that the overlap of its detected categories with $\mathcal{C}_{sel}$ is minimized, classes count is maximized, and the frame is as early as possible in *Balanced-TopK* sampling. Here, we explore each part effect on these components. Table 8 reports the ablation results of the three components in our proposed *Balanced-TopK* strategy: minimizing category overlap with the selected pool ($\mathcal{C}_{sel}$ Min.), maximizing the number of detected classes (Count Max.), and selecting the earliest frame in case of ties (Time Early). The complete version of our method achieves the best average performance across all question types, validating the necessity of each design component. Removing the category overlap minimization (*w/o $\mathcal{C}_{sel}$ Min.*) leads to the most significant performance drop, especially in multiple-choice tasks such as "Route Plan" and "Approach Order", indicating its crucial role in encouraging semantic diversity and avoiding redundant content across selected frames. Omitting the class count maximization (*w/o Count Max.*) also degrades performance, particularly in object-centric questions like "Obj. Count" and "Rel. Dist.", demonstrating that favoring frames with more detected objects helps maximize information gain per frame. Removing the temporal prioritization (*w/o Time Early*) slightly affects overall performance, with minor impacts across all metrics. This suggests that while encouraging earlier frame selection helps improve temporal coherence and avoids delayed keyframe concentration, it is relatively less critical than the other two components.

---

[3]https://github.com/ultralytics/ultralytics

Table 9: Ablation studies of each visual trajectories of Motion Reconstruction.

| Visual Trajectories | Avg. | Obj. Count | Abs. Dist. | Obj. Size | Room Size | Rel. Dist. | Rel. Dir. | Route Plan | Appr. Order |
|---|---|---|---|---|---|---|---|---|---|
| | | | Numerical Answer | | | | Multiple-Choice Answer | | |
| *w/o* $\mathbf{P}_{BEV}$ | 43.1 | 65.4 | 33.1 | 48.3 | 45.4 | 45.9 | 41.0 | 29.9 | 36.4 |
| *w/o* $\mathbf{P}_{3D}$ | 42.8 | 65.4 | 33.3 | 47.9 | 44.9 | 45.9 | 40.7 | 27.8 | 36.4 |

**Detector Choices.** Based on the ablation studies presented in Table 10, it is evident that selecting the optimal detector and sampling interval $N$ requires balancing detection accuracy and computational efficiency. Among the tested YOLO variants, YOLOv8N consistently achieves competitive average scores across numerical and multiple-choice question categories while maintaining the lowest inference time, especially at $N = 4$, where it balances a relatively high average accuracy of 43.2% with only 82 seconds processing time per video. Although larger models like YOLOv8S and YOLOv11S occasionally reach similar or slightly higher accuracy in some metrics, their time costs are significantly higher—often an order of magnitude more—making them less practical for real-time or resource-constrained scenarios. Notably, increasing the sampling interval $N$ generally reduces computation time but can cause minor fluctuations in accuracy, with $N = 4$ emerging as an optimal trade-off that avoids excessive temporal sparsity while reducing inference overhead. Moreover, while YOLOv11N shows marginal improvements over YOLOv8N in some tasks, its time cost nearly doubles. Therefore, our final choice of YOLOv8N with $N = 4$ reflects a well-justified compromise: it offers sufficient detection performance across diverse evaluation criteria without sacrificing speed, enabling efficient processing in practical deployment. This reinforces the insight that bigger and more complex models are not always better, and careful tuning of sampling intervals combined with lightweight detectors can yield a more balanced, efficient system.

Table 10: Ablation studies of different sample interval $N$ settings and detector choices. "Time(s)" denotes the average time consuming on processing videos from VSI-BENCH. "N" denotes the tiny version and "S" denotes the small version.

| Detector | $N$ | Time(s) | Avg. | Obj. Count | Abs. Dist. | Obj. Size | Room Size | Rel. Dist. | Rel. Dir. | Route Plan | Appr. Order |
|---|---|---|---|---|---|---|---|---|---|---|---|
| | | | | | Numerical Answer | | | | Multiple-Choice Answer | | |
| YOLOv8N | 1 | 410 | 42.9 | 66.2 | 31.7 | 48.2 | 47.3 | 42.8 | 42.0 | 30.4 | 33.8 |
| | 2 | 227 | 43.1 | 65.2 | 31.6 | 46.8 | 47.5 | 41.5 | 40.7 | 30.9 | 40.8 |
| | 3 | 159 | 42.3 | 66.4 | 32.1 | 46.6 | 48.1 | 40.8 | 39.8 | 30.9 | 34.0 |
| | 4 | 82 | 43.2 | 65.2 | 32.9 | 46.9 | 46.7 | 45.9 | 40.3 | 30.4 | 37.4 |
| | 8 | 63 | 42.8 | 66.5 | 33.6 | 46.0 | 46.7 | 41.7 | 41.5 | 30.9 | 35.9 |
| | 12 | 53 | 42.5 | 66.4 | 32.6 | 45.9 | 41.7 | 46.5 | 39.9 | 28.9 | 38.8 |
| YOLOv8S | 1 | 1989 | 42.4 | 66.2 | 31.6 | 46.9 | 46.4 | 39.4 | 41.8 | 29.9 | 37.2 |
| | 2 | 997 | 41.7 | 66.0 | 30.5 | 47.1 | 46.3 | 40.0 | 40.8 | 27.8 | 35.3 |
| | 3 | 518 | 43.2 | 66.9 | 31.3 | 47.3 | 48.4 | 43.8 | 40.6 | 29.4 | 38.3 |
| | 4 | 414 | 43.1 | 66.5 | 31.8 | 46.1 | 48.1 | 41.5 | 39.1 | 29.4 | 42.1 |
| | 8 | 236 | 41.9 | 64.6 | 29.2 | 46.1 | 43.9 | 37.2 | 41.2 | 30.4 | 43.2 |
| | 12 | 132 | 41.5 | 65.7 | 32.6 | 46.3 | 43.6 | 39.6 | 40.0 | 28.9 | 35.3 |
| YOLOv11N | 1 | 397 | 42.2 | 66.4 | 32.6 | 47.1 | 45.8 | 41.0 | 41.0 | 29.9 | 34.0 |
| | 2 | 219 | 42.7 | 66.6 | 32.1 | 46.4 | 46.8 | 40.1 | 39.2 | 31.9 | 39.0 |
| | 3 | 144 | 42.6 | 66.5 | 31.2 | 46.9 | 49.2 | 41.3 | 40.3 | 30.4 | 35.4 |
| | 4 | 75 | 43.2 | 66.2 | 34.0 | 45.5 | 48.0 | 42.1 | 41.0 | 31.4 | 37.4 |
| | 8 | 58 | 42.4 | 66.9 | 31.4 | 38.7 | 45.8 | 45.8 | 33.0 | 40.1 | 38.5 |
| | 12 | 47 | 42.5 | 66.4 | 32.6 | 46.5 | 41.7 | 46.5 | 28.9 | 39.9 | 38.8 |
| YOLOv11S | 1 | 1763 | 43.4 | 67.4 | 32.6 | 43.2 | 48.1 | 48.7 | 30.9 | 48.1 | 37.5 |
| | 2 | 940 | 43.0 | 67.1 | 32.3 | 43.5 | 47.0 | 47.2 | 28.9 | 40.8 | 37.2 |
| | 3 | 554 | 42.8 | 67.8 | 32.5 | 42.4 | 46.4 | 48.1 | 28.9 | 42.4 | 34.5 |
| | 4 | 397 | 42.4 | 66.8 | 32.5 | 41.5 | 46.4 | 48.9 | 30.9 | 39.7 | 32.5 |
| | 8 | 254 | 42.2 | 66.7 | 33.0 | 41.8 | 46.6 | 47.8 | 29.4 | 40.7 | 31.9 |
| | 12 | 113 | 41.8 | 66.5 | 33.2 | 40.7 | 46.2 | 45.2 | 28.4 | 40.4 | 34.1 |

**Keyframe Number Analysis.** We further investigate the impact of varying the number of keyframes $\mathcal{K}$ on model performance. As shown in Table 11, SEE&TREK consistently improves spatial understanding across all keyframe settings compared to the baseline INTERNVL3-8B. The most notable gain of +3.0% is observed at $\mathcal{K} = 8$, demonstrating that our method is highly effective even under sparse temporal input. Interestingly, as $\mathcal{K}$ increases, the performance improvement becomes marginal, with the gain reduced to +0.6% at $\mathcal{K} = 32$. This diminishing return can be attributed to the saturation of spatial information in densely sampled frames: when adjacent keyframes are visually redundant, they provide limited additional cues for the model to reason over spatial relationships. In contrast, our semantic richness sampling strategy is designed to select frames with diverse scene structures and salient spatial cues. Therefore, even a small number of informative frames can sufficiently support spatial reasoning, while denser sampling introduces redundancy without significantly enhancing spa-

tial understanding. This further validates the efficiency and robustness of SEE&TREK in leveraging a compact, semantically diverse set of frames to enhance MLLMs.

Table 11: Ablation studies of different keyframe number $\mathcal{K}$ setting. Method "-" denotes the baseline MLLM INTERNVL3-8B.

| $\mathcal{K}$ | Method | Avg. | Obj. Count | Abs. Dist. | Obj. Size | Room Size | Rel. Dist. | Rel. Dir. | Route Plan | Appr. Order |
| --- | --- | --- | --- | --- | --- | --- | --- | --- | --- | --- |
| | | | | Numerical Answer | | | | Multiple-Choice Answer | | |
| 8 | - | 40.2 | 67.8 | 32.2 | 44.5 | 41.8 | 42.8 | 37.7 | 26.3 | 28.3 |
| | +SEE&TREK | $43.2_{+3.0\%}$ | $65.2_{-2.6\%}$ | $32.9_{+0.7\%}$ | $46.9_{+2.4\%}$ | $46.7_{+4.9\%}$ | $45.9_{+3.1\%}$ | $40.2_{+2.5\%}$ | $30.4_{+4.1\%}$ | $37.4_{+9.1\%}$ |
| 12 | - | 41.8 | 67.8 | 32.2 | 44.5 | 41.8 | 42.8 | 37.7 | 26.3 | 28.3 |
| | +SEE&TREK | $43.6_{+1.8\%}$ | $67.8_{+0.0\%}$ | $34.0_{+1.8\%}$ | $43.4_{-1.1\%}$ | $45.0_{+3.2\%}$ | $43.9_{+1.1\%}$ | $39.7_{+2.0\%}$ | $26.3_{+0.0\%}$ | $34.3_{+6.0\%}$ |
| 16 | - | 42.4 | 69.3 | 33.9 | 43.1 | 47.3 | 44.8 | 38.7 | 26.8 | 35.3 |
| | +SEE&TREK | $44.0_{+1.6\%}$ | $67.3_{-2.0\%}$ | $34.6_{+0.7\%}$ | $48.1_{+5.0\%}$ | $47.9_{+0.6\%}$ | $44.9_{+0.1\%}$ | $39.7_{+1.0\%}$ | $29.4_{+2.6\%}$ | $39.6_{+4.3\%}$ |
| 24 | - | 43.2 | 69.6 | 35.8 | 44.3 | 45.9 | 49.2 | 39.1 | 27.8 | 33.7 |
| | +SEE&TREK | $44.1_{+0.9\%}$ | $67.5_{-2.1\%}$ | $34.5_{-1.3\%}$ | $46.6_{+2.3\%}$ | $49.0_{+3.1\%}$ | $47.3_{-1.9\%}$ | $39.0_{-0.1\%}$ | $30.4_{+2.6\%}$ | $38.7_{+5.0\%}$ |
| 32 | - | 43.1 | 69.3 | 34.9 | 41.6 | 49.4 | 46.9 | 38.9 | 29.9 | 33.9 |
| | +SEE&TREK | $43.7_{+0.6\%}$ | $68.5_{-0.8\%}$ | $34.3_{-0.6\%}$ | $46.1_{+4.5\%}$ | $48.1_{-1.3\%}$ | $45.6_{-1.3\%}$ | $41.5_{+2.6\%}$ | $28.4_{-1.5\%}$ | $36.7_{+2.8\%}$ |

**Motion Trajectory.** As mentioned in Section 3.2, we also investigate performance impact of different part of input visual trajectories containing BEV $\mathbf{P}_{BEV}$ and 3D figures $\mathbf{P}_{3D}$. Table 9 presents the ablation results evaluating the contribution of different visual trajectory components—BEV projections ($\mathbf{P}_{BEV}$) and 3D spatial figures ($\mathbf{P}_{3D}$) to the overall performance. Removing either component leads to a noticeable drop in average accuracy, confirming the complementary value of both trajectory views in enhancing spatial understanding. Specifically, excluding $\mathbf{P}_{BEV}$ results in slightly better performance than removing $\mathbf{P}_{3D}$, particularly in tasks requiring high-level route planning and temporal ordering, suggesting that BEV provides a more intuitive overview of the navigation path and spatial layout. Conversely, the 3D visualization offers critical depth and geometric cues, as evidenced by its impact on fine-grained spatial tasks such as "Obj. Size" and "Room Size." The results demonstrate that both $\mathbf{P}_{BEV}$ and $\mathbf{P}_{3D}$ are indispensable for supplying diverse and complementary motion context to the MLLM, enabling more robust reasoning across spatial tasks.

**Camera Intrinsics.** The fixed KITTI intrinsics are used solely within the visual odometry (VO) stage to convert 2D correspondences into relative camera motion. Our prompting design explicitly frames all motion cues in terms of relative distances and orientations, without providing any metric calibration to the MLLM. we include results using normalized, random and identity intrinsics matrices in the VO stage to demonstrate that the performance gains persist even without any meaningful camera calibration. Here, we still adopt the INTERNVL3-8B + SEE&TREK as the baseline and evaluate on the VSI-BENCH as shown in Table 12. The results in the table demonstrate that replacing the original KITTI intrinsics matrix with normalized, random, or identity matrices yields comparable performance across all spatial reasoning sub-tasks on VSI-BENCH. The minimal differences in average scores and individual metrics indicate that our method's effectiveness is not dependent on the accuracy or specific values of the camera intrinsics matrix. This supports our claim that the spatial improvements stem primarily from the relative motion cues encoded by our VO and prompting strategy, rather than any geometric advantage conferred by precise intrinsic calibration. Consequently, our framework maintains robustness and fairness even when intrinsic parameters are unavailable or approximate.

Table 12: Comparison results of different camera intrinsics setting $\mathbf{K}$.

| $\mathbf{K}$ | Avg. | Obj. Count | Abs. Dist. | Obj. Size | Room Size | Rel. Dist. | Rel. Dir. | Route Plan | Appr. Order |
| --- | --- | --- | --- | --- | --- | --- | --- | --- | --- |
| | | | Numerical Answer | | | | Multiple-Choice Answer | | |
| Original $\mathbf{K}$ [53] | 43.2 | 65.2 | 32.9 | 46.9 | 46.7 | 45.9 | 40.2 | 30.4 | 37.4 |
| Normalized $\mathbf{K}$ [54] | 43.0 | 65.2 | 32.6 | 46.5 | 47.2 | 46.8 | 39.0 | 28.9 | 37.5 |
| Random | 43.1 | 65.5 | 32.8 | 46.2 | 46.7 | 46.0 | 39.5 | 30.4 | 37.4 |
| Identity | 43.2 | 65.8 | 32.0 | 46.1 | 47.0 | 46.2 | 39.0 | 32.0 | 38.1 |

## B.6 Qualitative results

We give more illustration about how **SEE&TREK** impacts the baseline inference as shown in Figure 6/7/8. We also utilize the purple circle to highlight the objects mentioned in the question. For instance, in Figure 7(e), we observe that **SEE&TREK** can inspire more spatial reasoning in the baseline, such as more rational Chain of Thoughts (CoT) by incorporating more meaningful spatial information, leading to more accurate relative distance estimation. Besides, we also can conclude that the motion visual trajectories can improve the accuracy about the size estimation like Figure 6(b) like "given the

Table 13: Comparison results of different visual spatial methods containing SoM[53], CC [54] and our **SEE&TREK**. It shows that **SEE&TREK** obtains the SOTA performance in boosting the MLLMs spatial understanding.

| Method | Avg. | Obj. Count | Abs. Dist. | Obj. Size | Room Size | Rel. Dist. | Rel. Dir. | Route Plan | Appr. Order |
|--------|------|-----------|-----------|-----------|-----------|-----------|-----------|------------|-------------|
| | | **Numerical Answer** | | | | **Multiple-Choice Answer** | | | |
| SoM [53] | 41.3 | 64.5 | 29.6 | 44.9 | 43.1 | 36.8 | 41.1 | 27.3 | 43.7 |
| CC [54] | 40.9 | 64.4 | 30.0 | 43.8 | 38.3 | 39.1 | 41.6 | 27.8 | 42.2 |
| **SEE&TREK** | 43.2 | 65.2 | 32.9 | 46.9 | 46.7 | 45.9 | 40.2 | 30.4 | 37.4 |

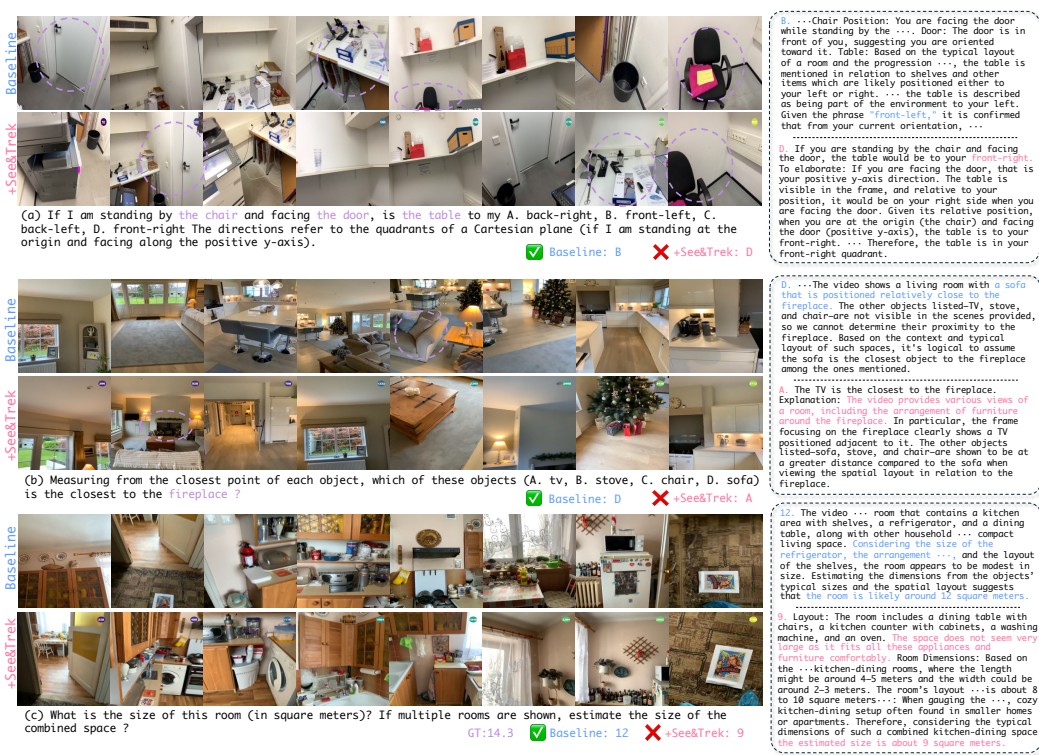

Figure 9: Illustration of several failure cases of **SEE&TREK** compared to the baseline. It contains three spatial tasks: (a) route plan (b) object relative distance (c) room size.

visual cues." Furthermore, we can get a conclusion that the Maximum Semantic Richness Sampling (MSRS) can boost the richness of extracted frames from the given video as shown in Figure 6(c) and 7(d,e,f). Overall, the proposed **SEE&TREK** nightlight its advantages in modeling egocentric movement and temporal coherence.

## B.7 Failure Cases

As shown in Figure 9, we also investigate the failure cases generate from **SEE&TREK** with empirical study. We first conclude that our baseline model maybe confused by the provided spatial information in some domains. For instance, in Figure 9(a) or 9(b), we observe that even though existing all objects like door or fireplace from the query in the selected frames, the answer generated from **SEE&TREK** is wrong. Besides, it also get the conclusion that choosing GPU-free YOLO as the perception model still does not have strong generalization ability and performs poorly in multi object and multi class scenes like Figure 9(c), which lacks more goals compared to the baseline. It motivates us to utilize more powerful dense perception model to extract higher semantic richness frames from the given spatial videos.

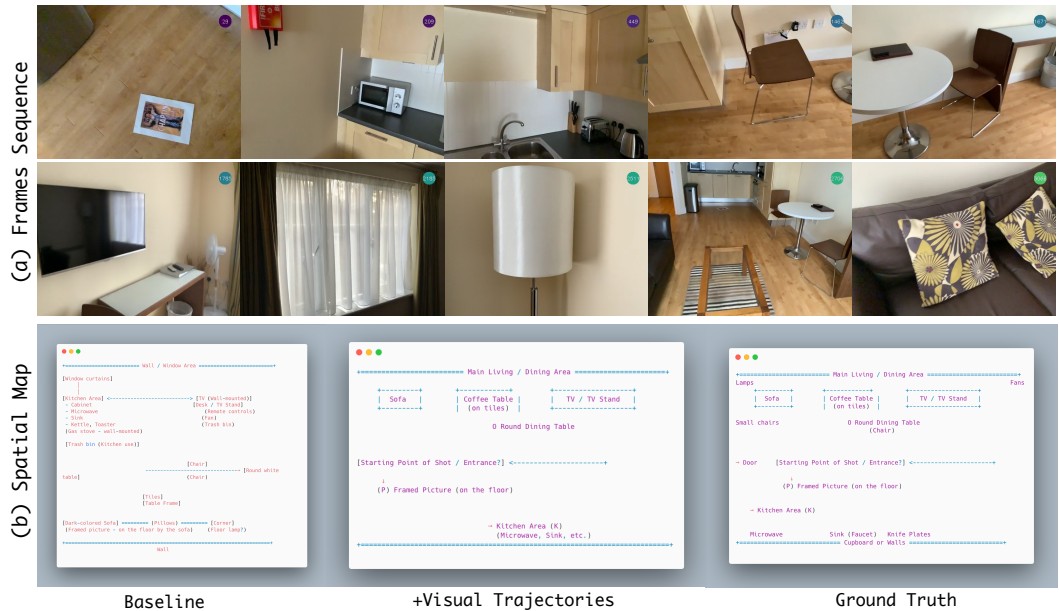

Figure 10: Illustration of the implicit impact of visual trajectories from SEE&TREK for MLLMs internal inference. Here, we utilize GEMINI 2.5 PRO as our baseline. It demonstrates that visual trajectories can help MLLMs construct better implicit spatial map, which improve different spatial task performance comprehensively.

## B.8   More Prompting Comparison

SEE&TREK is the first training&gpu-free method to improve the spatial understanding ability of MLLMs. Served as the prompting methods [55, 56, 57, 58], we also explore other current visual prompting method [59, 53, 54] in other domains for comparison. We notice the 3DAXISPROMPT [59] also can boost MLLMs spatial understanding which leverages the 3D coordinate axis and masks generated from the Segment Anything Model (SAM) [60] to provide explicit geometric priors. Due to its lack of open-source code which makes it hard to review, we only discuss it here. Then, we leverage Set of Mask (SOM) [53] and Coarse Correspondence (CC) [54], which enable the identification of objects—either with masks or numeric labels, by utilizing the capabilities of current vision-language models. Similar from the setting [53, 54], we also sample uniformly 8 frames from given videos and conduct these method to process. We still follow the above-mentioned setting to evaluate, *e.g.* INTERNVL3-8B as baseline. As shown in Table 13, our proposed SEE&TREK consistently outperforms existing visual prompting methods such as SOM and CC across all spatial reasoning tasks. While SOM and CC primarily rely on static masks or coarse positional labels to identify object regions, they offer limited support for modeling inter-object relationships or capturing scene-level spatial structures. In contrast, SEE&TREK leverages two key principles—visual diversity and motion reconstruction—to provide richer and more structured spatial cues. Specifically, the Maximum Semantic Richness Sampling strategy ensures that the input frames encapsulate diverse spatial layouts and semantic contexts, while the simulated visual trajectories explicitly encode relative positions and temporal continuity, which are essential for complex spatial reasoning tasks. Unlike prior approaches that depend on task-specific annotations or domain priors, SEE&TREK operates in a fully training- and GPU-free manner, making it both lightweight and generalizable. This design enables MLLMs to develop a deeper and more coherent spatial understanding, as reflected in the consistent improvements across all evaluated tasks.

## C   Discussion: Trajectories Help MLLMs Construct Implicit Spatial Map

As discussed in earlier sections, SEE&TREK is designed to enhance the spatial understanding capabilities of MLLMs under limited visual input by preserving both spatial relationships and temporal coherence. While Maximum Semantic Richness Sampling effectively improves visual

diversity, the influence of motion reconstruction on the model's internal inference processes remains less understood. To investigate this, we move beyond explicit reasoning analyses (*e.g.*, Figure 3) and propose a more challenging task: generating structured spatial maps in text form. This setting offers a more direct and interpretable view into the model's implicit spatial reasoning and internal representational mechanisms. As shown in Figure 10, the baseline model exhibits limited spatial coherence—although relevant entities are identified, their placements are often disorganized and misaligned with the actual room layout. In contrast, the model equipped with visual trajectories produces spatial maps that more closely resemble the ground truth. These outputs demonstrate more accurate object positioning (*e.g.*, correctly situating the kitchen area and entrance) and better structural grouping (*e.g.*, aligning the coffee table with the sofa and TV stand). The observed improvement underscores the critical role of motion-guided trajectories in reinforcing spatial continuity and layout reasoning. By temporally linking semantically rich keyframes, visual trajectories provide the model with stronger contextual cues, helping it infer object relationships and transitions across frames. In essence, these trajectories act as a soft inductive prior, enabling MLLMs to construct spatially consistent internal representations from fragmented visual observations.

# D  Discussion: When SEE&TREK Meet Powerful Perception Models

This paper mainly proposes a general and training-free framework that explicitly incorporates semantic-rich frame sampling and motion reconstruction to enhance spatial reasoning. To validate this idea in a lightweight and accessible manner, we implement a minimal working solution using YOLOV8-TINY and manual VO module, both of which are GPU-free and efficient. This implementation demonstrates that even under constrained settings, our method can deliver consistent performance gains. Importantly, our framework is highly extensible. The perception and motion modules can be readily replaced with stronger alternatives according to different scenarios. Here, we conduct experiments on the VSI-BENCH involving the replacement of various alternatives, such as substituting the detection component with VLM-based Models (GROUNDING DINO [61] and YOLO-WORLD [62]) as shown in Table 14. Note that we utilize the INTERNVL3-8B+SEE&TREK as our baseline.

Table 14: Comparison results of utilizing more powerful perception models.

| Method | Time(s) | GPU Occ.(MB) | Avg. | Obj. Count | Abs. Dist. | Obj. Size | Room Size | Rel. Dist. | Rel. Dir. | Route Plan | Appr. Order |
|---|---|---|---|---|---|---|---|---|---|---|---|
| | | | | | Numerical Answer | | | | Multiple-Choice Answer | | |
| Baseline | 82 | 0 | 43.2 | 65.2 | 32.9 | 46.9 | 46.7 | 45.9 | 40.3 | 30.4 | 37.4 |
| GROUNDING DINO [61] | 157 | 5430 | 43.8 | 66.3 | 32.3 | 46.7 | 47.1 | 47.4 | 41.4 | 31.3 | 38.2 |
| YOLO-WORLD [62] | 107 | 1740 | 43.5 | 66.2 | 31.8 | 46.2 | 49.5 | 44.2 | 38.9 | 30.9 | 37.4 |

# E  Acknowledgment

This work was supported in part by the National Key R&D Program of China (Grant No.2023YFF0725001), in part by the National Natural Science Foundation of China (Grant No.92370204), in part by the guangdong Basic and Applied Basic Research Foundation (Grant No.2023B1515120057), in part by the Education Bureau of Guangzhou.

# NeurIPS Paper Checklist

1. **Claims**

   Question: Do the main claims made in the abstract and introduction accurately reflect the paper's contributions and scope?

   Answer: [Yes]

   Justification: The abstract and introduction clearly state the core contributions and scope of the paper. Specifically, the claims focus on enhancing spatial understanding in Multimodal Large Language Models (MLLMs) under purely visual constraints, which is a well-motivated and underexplored area. The abstract introduces the proposed **SEE&TREK** framework, emphasizing its two main components—semantic keyframe sampling and motion reconstruction—both of which are described in terms of their purpose and implementation. Furthermore, the claim of being training-free and easily integrable is directly supported by the method's design and empirical results. The reported improvements on benchmark datasets (VSI-BENCH and STI-BENCH) further substantiate the claimed performance gains.

   Guidelines:
   - The answer NA means that the abstract and introduction do not include the claims made in the paper.
   - The abstract and/or introduction should clearly state the claims made, including the contributions made in the paper and important assumptions and limitations. A No or NA answer to this question will not be perceived well by the reviewers.
   - The claims made should match theoretical and experimental results, and reflect how much the results can be expected to generalize to other settings.
   - It is fine to include aspirational goals as motivation as long as it is clear that these goals are not attained by the paper.

2. **Limitations**

   Question: Does the paper discuss the limitations of the work performed by the authors?

   Answer: [Yes]

   Justification: The paper includes a dedicated "Limitations" section in the appendix, which clearly acknowledges the constraints of the proposed **SEE&TREK** framework. The authors discuss how the performance of the method depends on the external perception model used for keyframe extraction and motion reconstruction, highlighting that improvements are bounded by the semantic richness and capability of these models. Additionally, they note that while the framework is designed to be GPU- and training-free, relaxing these constraints in the future could lead to further performance gains. This discussion reflects an awareness of the assumptions and practical limitations of the current design, and shows consideration for how the method might generalize or be extended in future work.

   Guidelines:
   - The answer NA means that the paper has no limitation while the answer No means that the paper has limitations, but those are not discussed in the paper.
   - The authors are encouraged to create a separate "Limitations" section in their paper.
   - The paper should point out any strong assumptions and how robust the results are to violations of these assumptions (e.g., independence assumptions, noiseless settings, model well-specification, asymptotic approximations only holding locally). The authors should reflect on how these assumptions might be violated in practice and what the implications would be.
   - The authors should reflect on the scope of the claims made, e.g., if the approach was only tested on a few datasets or with a few runs. In general, empirical results often depend on implicit assumptions, which should be articulated.
   - The authors should reflect on the factors that influence the performance of the approach. For example, a facial recognition algorithm may perform poorly when image resolution is low or images are taken in low lighting. Or a speech-to-text system might not be used reliably to provide closed captions for online lectures because it fails to handle technical jargon.
   - The authors should discuss the computational efficiency of the proposed algorithms and how they scale with dataset size.

- If applicable, the authors should discuss possible limitations of their approach to address problems of privacy and fairness.
- While the authors might fear that complete honesty about limitations might be used by reviewers as grounds for rejection, a worse outcome might be that reviewers discover limitations that aren't acknowledged in the paper. The authors should use their best judgment and recognize that individual actions in favor of transparency play an important role in developing norms that preserve the integrity of the community. Reviewers will be specifically instructed to not penalize honesty concerning limitations.

3. **Theory assumptions and proofs**

   Question: For each theoretical result, does the paper provide the full set of assumptions and a complete (and correct) proof?

   Answer: [NA]

   Justification: The paper does not include any theoretical results, theorems, or formal proofs. Its contributions are methodological and experimental, focusing on the design of a training-free prompting framework and empirical validation on spatial reasoning benchmarks.

   Guidelines:
   - The answer NA means that the paper does not include theoretical results.
   - All the theorems, formulas, and proofs in the paper should be numbered and cross-referenced.
   - All assumptions should be clearly stated or referenced in the statement of any theorems.
   - The proofs can either appear in the main paper or the supplemental material, but if they appear in the supplemental material, the authors are encouraged to provide a short proof sketch to provide intuition.
   - Inversely, any informal proof provided in the core of the paper should be complemented by formal proofs provided in appendix or supplemental material.
   - Theorems and Lemmas that the proof relies upon should be properly referenced.

4. **Experimental result reproducibility**

   Question: Does the paper fully disclose all the information needed to reproduce the main experimental results of the paper to the extent that it affects the main claims and/or conclusions of the paper (regardless of whether the code and data are provided or not)?

   Answer: [Yes]

   Justification: Our paper have explained the detailed experiment setting in the appendix like "Implementation" and the hardware environment. This information is sufficient for others to estimate computer requirements and reproduce the experiments.

   Guidelines:
   - The answer NA means that the paper does not include experiments.
   - If the paper includes experiments, a No answer to this question will not be perceived well by the reviewers: Making the paper reproducible is important, regardless of whether the code and data are provided or not.
   - If the contribution is a dataset and/or model, the authors should describe the steps taken to make their results reproducible or verifiable.
   - Depending on the contribution, reproducibility can be accomplished in various ways. For example, if the contribution is a novel architecture, describing the architecture fully might suffice, or if the contribution is a specific model and empirical evaluation, it may be necessary to either make it possible for others to replicate the model with the same dataset, or provide access to the model. In general. releasing code and data is often one good way to accomplish this, but reproducibility can also be provided via detailed instructions for how to replicate the results, access to a hosted model (e.g., in the case of a large language model), releasing of a model checkpoint, or other means that are appropriate to the research performed.
   - While NeurIPS does not require releasing code, the conference does require all submissions to provide some reasonable avenue for reproducibility, which may depend on the nature of the contribution. For example
     (a) If the contribution is primarily a new algorithm, the paper should make it clear how to reproduce that algorithm.

(b) If the contribution is primarily a new model architecture, the paper should describe the architecture clearly and fully.

(c) If the contribution is a new model (e.g., a large language model), then there should either be a way to access this model for reproducing the results or a way to reproduce the model (e.g., with an open-source dataset or instructions for how to construct the dataset).

(d) We recognize that reproducibility may be tricky in some cases, in which case authors are welcome to describe the particular way they provide for reproducibility. In the case of closed-source models, it may be that access to the model is limited in some way (e.g., to registered users), but it should be possible for other researchers to have some path to reproducing or verifying the results.

5. **Open access to data and code**

Question: Does the paper provide open access to the data and code, with sufficient instructions to faithfully reproduce the main experimental results, as described in supplemental material?

Answer: [Yes]

Justification: Our paper will submit the code in a single zip file along with additional supplementary material.

Guidelines:
- The answer NA means that paper does not include experiments requiring code.
- Please see the NeurIPS code and data submission guidelines (`https://nips.cc/public/guides/CodeSubmissionPolicy`) for more details.
- While we encourage the release of code and data, we understand that this might not be possible, so "No" is an acceptable answer. Papers cannot be rejected simply for not including code, unless this is central to the contribution (e.g., for a new open-source benchmark).
- The instructions should contain the exact command and environment needed to run to reproduce the results. See the NeurIPS code and data submission guidelines (`https://nips.cc/public/guides/CodeSubmissionPolicy`) for more details.
- The authors should provide instructions on data access and preparation, including how to access the raw data, preprocessed data, intermediate data, and generated data, etc.
- The authors should provide scripts to reproduce all experimental results for the new proposed method and baselines. If only a subset of experiments are reproducible, they should state which ones are omitted from the script and why.
- At submission time, to preserve anonymity, the authors should release anonymized versions (if applicable).
- Providing as much information as possible in supplemental material (appended to the paper) is recommended, but including URLs to data and code is permitted.

6. **Experimental setting/details**

Question: Does the paper specify all the training and test details (e.g., data splits, hyper-parameters, how they were chosen, type of optimizer, etc.) necessary to understand the results?

Answer: [Yes]

Justification: The paper provides sufficient experimental details to allow readers to understand and reproduce the results. It clearly specifies the benchmarks used (VSI-BENCH and STI-BENCH), the MLLMs evaluated, and the setup of the inference-only prompting framework.

Guidelines:
- The answer NA means that the paper does not include experiments.
- The experimental setting should be presented in the core of the paper to a level of detail that is necessary to appreciate the results and make sense of them.
- The full details can be provided either with the code, in appendix, or as supplemental material.

7. **Experiment statistical significance**

Question: Does the paper report error bars suitably and correctly defined or other appropriate information about the statistical significance of the experiments?

Answer: [No]

Justification: While the paper presents empirical results that support the effectiveness of the proposed **SEE&TREK** framework, it does not report error bars, confidence intervals, or statistical significance tests. As a result, it is unclear how sensitive the reported performance improvements are to factors such as random sampling or variability in model inference. However, our evaluation set the random seed to a fixed number to guarantee the re-productivity of our paper. Including such statistical analysis would help better assess the robustness and reliability of the findings.

Guidelines:
- The answer NA means that the paper does not include experiments.
- The authors should answer "Yes" if the results are accompanied by error bars, confidence intervals, or statistical significance tests, at least for the experiments that support the main claims of the paper.
- The factors of variability that the error bars are capturing should be clearly stated (for example, train/test split, initialization, random drawing of some parameter, or overall run with given experimental conditions).
- The method for calculating the error bars should be explained (closed form formula, call to a library function, bootstrap, etc.)
- The assumptions made should be given (e.g., Normally distributed errors).
- It should be clear whether the error bar is the standard deviation or the standard error of the mean.
- It is OK to report 1-sigma error bars, but one should state it. The authors should preferably report a 2-sigma error bar than state that they have a 96% CI, if the hypothesis of Normality of errors is not verified.
- For asymmetric distributions, the authors should be careful not to show in tables or figures symmetric error bars that would yield results that are out of range (e.g. negative error rates).
- If error bars are reported in tables or plots, The authors should explain in the text how they were calculated and reference the corresponding figures or tables in the text.

8. **Experiments compute resources**

Question: For each experiment, does the paper provide sufficient information on the computer resources (type of compute workers, memory, time of execution) needed to reproduce the experiments?

Answer: [Yes]

Justification: We have provided the detailed development environment such as hardware in the Appendix "Implementation" part. We also report the running time of our proposed method in the Paper Table 5.

Guidelines:
- The answer NA means that the paper does not include experiments.
- The paper should indicate the type of compute workers CPU or GPU, internal cluster, or cloud provider, including relevant memory and storage.
- The paper should provide the amount of compute required for each of the individual experimental runs as well as estimate the total compute.
- The paper should disclose whether the full research project required more compute than the experiments reported in the paper (e.g., preliminary or failed experiments that didn't make it into the paper).

9. **Code of ethics**

Question: Does the research conducted in the paper conform, in every respect, with the NeurIPS Code of Ethics `https://neurips.cc/public/EthicsGuidelines`?

Answer: [Yes]

Justification: The research adheres to the NeurIPS Code of Ethics. The proposed **SEE&TREK** framework is designed to improve the spatial reasoning capabilities of Multimodal Large Language Models without requiring additional training or sensitive data. The work relies solely on publicly available benchmarks and off-the-shelf models, without involving human subjects, private user data, or potentially harmful applications. Moreover, care has been taken to ensure that the framework is efficient and accessible (*e.g.*, GPU-free), aligning with the principles of responsible AI development and equitable research access.

Guidelines:
- The answer NA means that the authors have not reviewed the NeurIPS Code of Ethics.
- If the authors answer No, they should explain the special circumstances that require a deviation from the Code of Ethics.
- The authors should make sure to preserve anonymity (e.g., if there is a special consideration due to laws or regulations in their jurisdiction).

10. **Broader impacts**

Question: Does the paper discuss both potential positive societal impacts and negative societal impacts of the work performed?

Answer: [NA]

Justification: The paper does not explicitly discuss broader societal impacts. While the proposed framework focuses on foundational improvements to spatial reasoning in MLLMs and is not directly tied to specific applications, a discussion of both potential benefits and risks (*e.g.*, misuse in surveillance or failure in safety-critical systems) would strengthen the work's alignment with ethical and responsible AI practices.

Guidelines:
- The answer NA means that there is no societal impact of the work performed.
- If the authors answer NA or No, they should explain why their work has no societal impact or why the paper does not address societal impact.
- Examples of negative societal impacts include potential malicious or unintended uses (e.g., disinformation, generating fake profiles, surveillance), fairness considerations (e.g., deployment of technologies that could make decisions that unfairly impact specific groups), privacy considerations, and security considerations.
- The conference expects that many papers will be foundational research and not tied to particular applications, let alone deployments. However, if there is a direct path to any negative applications, the authors should point it out. For example, it is legitimate to point out that an improvement in the quality of generative models could be used to generate deepfakes for disinformation. On the other hand, it is not needed to point out that a generic algorithm for optimizing neural networks could enable people to train models that generate Deepfakes faster.
- The authors should consider possible harms that could arise when the technology is being used as intended and functioning correctly, harms that could arise when the technology is being used as intended but gives incorrect results, and harms following from (intentional or unintentional) misuse of the technology.
- If there are negative societal impacts, the authors could also discuss possible mitigation strategies (e.g., gated release of models, providing defenses in addition to attacks, mechanisms for monitoring misuse, mechanisms to monitor how a system learns from feedback over time, improving the efficiency and accessibility of ML).

11. **Safeguards**

Question: Does the paper describe safeguards that have been put in place for responsible release of data or models that have a high risk for misuse (e.g., pretrained language models, image generators, or scraped datasets)?

Answer: [NA]

Justification: The paper does not introduce or release any new pretrained models or scraped datasets that carry a high risk of misuse. Instead, it proposes a training-free prompting framework that leverages existing publicly available MLLMs and off-the-shelf perception models. As such, the research poses no new risks that would require specific safeguards for responsible release.

Guidelines:
- The answer NA means that the paper poses no such risks.
- Released models that have a high risk for misuse or dual-use should be released with necessary safeguards to allow for controlled use of the model, for example by requiring that users adhere to usage guidelines or restrictions to access the model or implementing safety filters.
- Datasets that have been scraped from the Internet could pose safety risks. The authors should describe how they avoided releasing unsafe images.
- We recognize that providing effective safeguards is challenging, and many papers do not require this, but we encourage authors to take this into account and make a best faith effort.

12. **Licenses for existing assets**

Question: Are the creators or original owners of assets (e.g., code, data, models), used in the paper, properly credited and are the license and terms of use explicitly mentioned and properly respected?

Answer: [Yes]

Justification: The paper makes use of existing publicly available models and datasets, including open-source MLLMs (*e.g.*, Qwen, LLaVA) and perception models (*e.g.*, YOLOv8). All assets used are properly credited through citations in the main text or appendix. The licenses of these assets—such as Apache 2.0 or MIT for code, and CC-BY or similar for datasets—are respected. For each asset, version information and URLs to official repositories are provided to ensure traceability and compliance with their terms of use. No proprietary or restricted data was employed in the study.

Guidelines:
- The answer NA means that the paper does not use existing assets.
- The authors should cite the original paper that produced the code package or dataset.
- The authors should state which version of the asset is used and, if possible, include a URL.
- The name of the license (e.g., CC-BY 4.0) should be included for each asset.
- For scraped data from a particular source (e.g., website), the copyright and terms of service of that source should be provided.
- If assets are released, the license, copyright information, and terms of use in the package should be provided. For popular datasets, `paperswithcode.com/datasets` has curated licenses for some datasets. Their licensing guide can help determine the license of a dataset.
- For existing datasets that are re-packaged, both the original license and the license of the derived asset (if it has changed) should be provided.
- If this information is not available online, the authors are encouraged to reach out to the asset's creators.

13. **New assets**

Question: Are new assets introduced in the paper well documented and is the documentation provided alongside the assets?

Answer: [NA]

Justification: The paper does not introduce or release any new assets such as datasets, models, or software. Instead, it builds on existing publicly available models and benchmarks, without modification or redistribution. As such, there is no need for additional documentation or licensing related to new assets.

Guidelines:
- The answer NA means that the paper does not release new assets.
- Researchers should communicate the details of the dataset/code/model as part of their submissions via structured templates. This includes details about training, license, limitations, etc.
- The paper should discuss whether and how consent was obtained from people whose asset is used.

- At submission time, remember to anonymize your assets (if applicable). You can either create an anonymized URL or include an anonymized zip file.

14. **Crowdsourcing and research with human subjects**

    Question: For crowdsourcing experiments and research with human subjects, does the paper include the full text of instructions given to participants and screenshots, if applicable, as well as details about compensation (if any)?

    Answer: [NA]

    Justification: The paper does not involve any crowdsourcing experiments or research with human subjects. All evaluations are conducted using existing benchmark datasets and pre-trained models. No new data was collected from human participants, and no human-subject interactions were involved in any stage of the research.

    Guidelines:
    - The answer NA means that the paper does not involve crowdsourcing nor research with human subjects.
    - Including this information in the supplemental material is fine, but if the main contribution of the paper involves human subjects, then as much detail as possible should be included in the main paper.
    - According to the NeurIPS Code of Ethics, workers involved in data collection, curation, or other labor should be paid at least the minimum wage in the country of the data collector.

15. **Institutional review board (IRB) approvals or equivalent for research with human subjects**

    Question: Does the paper describe potential risks incurred by study participants, whether such risks were disclosed to the subjects, and whether Institutional Review Board (IRB) approvals (or an equivalent approval/review based on the requirements of your country or institution) were obtained?

    Answer: [NA]

    Justification: The paper does not involve any research with human subjects or crowdsourcing. All experiments are conducted using publicly available datasets and models, and no human participants were involved at any stage of the research, thereby not requiring IRB or equivalent approval.

    Guidelines:
    - The answer NA means that the paper does not involve crowdsourcing nor research with human subjects.
    - Depending on the country in which research is conducted, IRB approval (or equivalent) may be required for any human subjects research. If you obtained IRB approval, you should clearly state this in the paper.
    - We recognize that the procedures for this may vary significantly between institutions and locations, and we expect authors to adhere to the NeurIPS Code of Ethics and the guidelines for their institution.
    - For initial submissions, do not include any information that would break anonymity (if applicable), such as the institution conducting the review.

16. **Declaration of LLM usage**

    Question: Does the paper describe the usage of LLMs if it is an important, original, or non-standard component of the core methods in this research? Note that if the LLM is used only for writing, editing, or formatting purposes and does not impact the core methodology, scientific rigorousness, or originality of the research, declaration is not required.

    Answer: [No]

    Justification: The paper uses large language models (LLMs) solely for writing, editing, and formatting purposes, without involving LLMs as an important, original, or non-standard component of the core methods or experiments. Therefore, the usage of LLMs does not impact the scientific rigor or originality of the research.

    Guidelines:

- The answer NA means that the core method development in this research does not involve LLMs as any important, original, or non-standard components.
- Please refer to our LLM policy (`https://neurips.cc/Conferences/2025/LLM`) for what should or should not be described.

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
