# OpenReview forum: "See&Trek: Training-Free Spatial Prompting for Multimodal Large Language Model"
_NeurIPS.cc/2025/Conference — NeurIPS 2025 poster_

### Official Review · Reviewer_Jfjm · 2025-06-23

**Clarity:** 2
**Significance:** 3
**Originality:** 3
**Rating:** 4
**Confidence:** 3

**Summary:**

This paper proposes SEE&TREK, a training-free spatial prompting framework for enhancing the spatial reasoning ability of Multimodal Large Language Models (MLLMs) under vision-only settings. The method addresses two major limitations of existing MLLMs in spatial tasks: visual homogeneity due to uniform frame sampling and unknown motion due to lack of temporal cues. To this end, SEE&TREK introduces two components: (1) Maximum Semantic Richness Sampling (MSRS), which selects visually diverse keyframes via a Balanced-TopK strategy using object detectors; and (2) Motion Reconstruction, which estimates camera motion through visual odometry and overlays spatiotemporal cues onto keyframes. These enhanced inputs are formatted into a spatially-aware prompt for MLLMs. Experiments on VSI-BENCH and STI-BENCH show improvements across several open-source MLLMs.

**Questions:**

see weakness.

**Ethical Concerns:**

["NO or VERY MINOR ethics concerns only"]

**Final Justification:**

After carefully considering the authors’ rebuttal and the subsequent discussions with the authors, other reviewers, and the Area Chair, I provide the following justification for my recommended score:

Issues Resolved:

The authors have adequately addressed my initial technical concerns and clarified key points regarding their methodology and experimental setup.

Remaining Issues:

The manuscript currently lacks a dedicated section that thoroughly discusses the relationship between this work, the strong baseline, and closely related prior work.

I will keep my score.

**Limitations:**

none.

**Quality:**

3

**Strengths And Weaknesses:**

Strengths

1. The paper addresses an important and underexplored challenge—improving spatial reasoning in vision-only MLLMs without training—which is both timely and practically relevant.

2. The method is training-free, requiring only CPU-based post-processing. This makes it deployable for lightweight scenarios and commercial APIs.

3. The paper introduces a way to inject both semantic richness and temporal motion cues into prompts, which is intuitive. The Balanced-TopK sampling and motion-aware frame augmentation are lightweight.

4. The method is evaluated across two spatial benchmarks and multiple model families, with some improvements. Ablation studies and runtime analysis support the design choices.


Weaknesses


1. The paper lacks discussion and citation of related “training-free” multimodal prompting approaches[1,2,3].

[1] Bridging the Dynamic Perception Gap: Training-Free Draft Chain-of-Thought for Dynamic Multimodal Spatial Reasoning.

[2] MLLMs Know Where to Look: Training-free Perception of Small Visual Details with Multimodal LLMs.

[3] Controlmllm: Training-free visual prompt learning for multimodal large language models.

2. Some components (e.g., Balanced-TopK vs. Temporal-TopK) yield marginal improvements, but the significance of individual modules is not always clear. For example, Table 3 and 4 show that improvements from each component are small and sometimes inconsistent across tasks.

3. In Table 1, SEE&TREK is evaluated on four baseline models, but in Table 2 (STI-BENCH), only two open-source models are reported. This inconsistency raises concerns about potential cherry-picking or underperformance on omitted models. The authors should clarify the reason for the missing entries and, ideally, provide complete results across all evaluated models for both benchmarks.


4. The method is claimed to be seamlessly integrated into MLLMs, but it is only validated on LLaVA, Qwen, InternVL, and Kimi, with no experiments on proprietary models like GPT-4V or Gemini despite them being listed in tables. The relative gain in strong models (e.g., Qwen2.5-VL-32B) is modest, raising questions about scalability and ceiling effect.

5. The method introduces several manually designed priors, which may overfit to specific types of tasks in VSI/STI-BENCH. It remains unclear how well this generalizes to other spatial reasoning tasks or open-world questions. This is the most important issue and also my primary concern.

6. The proposed method heavily relies on off-the-shelf models such as object detectors (e.g., YOLO) and others. The authors should provide ablation studies of these auxiliary models to better understand their necessity and influence on the overall performance.

7. The paper centers on video understanding. The authors should clarify that the scope is limited to video inputs and distinguish their approach from image-based spatial understanding tasks.

---

> ### Author Rebuttal · Authors · 2025-07-31
>
> We thank the reviewer for their thoughtful comments. We are glad that the reviewer appreciated the practical relevance and timeliness of addressing spatial reasoning in vision-only MLLMs, and the intuitive design supported by thorough evaluations. We now address your comments point by point below.
>
> 1. **Concern about Related Work Discussions.** These works have made important contributions to training-free visual prompting paradigms. Here, we compensate for these discussions with these works as follows: D2R [1] is a training-free prompting framework, which integrates textual CoT with corresponding visual drafts into MLLMs. ViCrop [2] leverages a MLLM’s own internal attention and gradients to automatically crop images, significantly improving its perceptual accuracy on small visual details that it would otherwise fail to recognize. ControlMLLM [3] enables MLLMs to follow specific visual prompts by optimizing a learnable latent variable at inference time to control the model's attention. In addition to these works, we also discuss some prompting works that are beneficial to enhancing spatial understanding ability in the appendix Section B.6., such as CC [4] or SoM [5].  Different from these works, See&Trek aims to enhance the spatial understanding of MLLMs under vision-only constraints in a training & GPU-free manner, which focuses on two core principles: increasing visual diversity and motion reconstruction.
>
> 2. **Concern about Each Component Performance.** We appreciate the reviewer’s careful inspection of our ablation results (Table 3/4/5 in main papers).
>
> - a. Compared to Temporal-TopK, Balanced-TopK focuses more on semantic richness, which selects the frame with the most classes within each temporal segment in a global view. Thus, Balanced-TopK can cover object classes in the video as much as possible. The diversity of the collected scenes helps MLLM understand the spatial relation between objects, namely, the spatial structure in the video. The experiment in Table 6 provides evidence of the hypothesis:  Balanced-TopK achieves consistent gains on several key spatial reasoning tasks compared to Temporal-TopK like 1) Relative Distance: +4.9% 2) Room Size: +1.2% 3) Absolute Distance: +1.4% 4) Relative Direction and Approximate Order: +0.5% each. However, Balanced-TopK will ignore the frame with fewer objects whose classes are already collected. If these objects are not in the previously collected frames, MLLM will never be aware of them. Thus, we observe a performance decrease in the metrics related to objects themselves, e.g., Object Count.
>
> - b. Each module is designed to target specific spatial reasoning capabilities, rather than boosting all tasks uniformly. we would like to emphasize that the design of our method explicitly targets different subtypes of spatial reasoning, and the ablation tables (Table 3 and 4) demonstrate this alignment:
>    - MSRS focuses on increasing semantic diversity, which is reflected in its gains in Approximate Order (+3.6%) and Relative Direction (+6.6%)
>    - Motion Reconstruction enhances the model’s understanding of trajectory and transitions, leading to significant improvements in Route Plan (+2.5%) and Approaching Order (+14.7%).
>    - Spatiotemporal Encoding and Point Prompts aim to connect visual diversity component and motion construction, offering complementary spatiotemporal cues, showing targeted gains in Abs. Distance (+1.9 \~ 2.2%), Relative Distance (+1.7~2.0%), and Route Plan (+4.1%). It demonstrates that these design can aid in temporal alignment and enhance geometric grounding.
>    - The ablation study of the internal component design and the corresponding experiments can also be found in Appendix like Table 7-10. We hope that this additional experimental information can resolve your doubts.
>
> 3. **Concern about More Baseline Models.** We thank the reviewer for pointing out the inconsistency between the number of models reported in Table 1 (VSI-BENCH) and Table 2 (STI-BENCH). We acknowledge that in our original submission, only two open-source models were reported on STI-BENCH due to early-stage computational resource constraints. This was not intended as cherry-picking but rather a reflection of practical limitations during initial benchmarking. To address this concern, we now conduct a comprehensive evaluation of some previously tested MLLMs on STI-BENCH and incorporated the complete results into the revised version. As shown in the updated Table below, SEE&TREK consistently improves performance across most models, with gains ranging from +1.0% to +1.4% on average.
>
> | Model | w/o SEE&TREK | w/ SEE&TREK | Gain (%) |
> | --- | --- | --- | --- |
> | LLaVA-OneVision-0.5B | 16.6 | 18.8 | +1.2% |
> | LLaVA-OneVision-7B | 31.8 | 33.2 | +1.4% |
> | Kimi-VL-A3B-Instruct | 37.2 | 38.3 | +1.1% |
>
> 4. **Concern about the Closed-source Models Performance.** We appreciate the reviewer’s feedback regarding the evaluation scope across MLLMs, including proprietary models. Here, we report results for proprietary models (GPT-4o, Gemini-2.0-Flash) on VSI-BENCH.
>
> | Method | Gains | Avg. | Obj. Count | Abs. Dist. | Obj. Size | Room Size | Rel. Dist. | Rel. Dir. | Route Plan | Appr. Order |
> | --- | --- | --- | --- | --- | --- | --- | --- | --- | --- | --- |
> | Gemini-2.0-Flash | - | 42.6 | 51.1 | 29.8 | 55.1 | 56.2 | 38.2 | 40.9 | 32.1 | 37.4 |
> | +**SEE&TREK** | +1.2% | 43.8 | 49.8 | 29.3 | 56.4 | 57.0 | 39.4 | 41.9 | 35.4 | 41.8 |
> | GPT-4o | - | 34.0 | 46.2 | 5.3 | 43.8 | 38.2 | 37.0 | 41.3 | 31.5 | 28.5 |
> | +**SEE&TREK** | +1.4% | 35.4 | 48.5 | 5.0 | 45.1 | 40.6 | 39.0 | 43.2 | 33.5 | 31.5 |
>
> As shown above, these commercial engines benefit from our SEE&TREK framework across various spatial tasks, reinforcing its generality and practical effectiveness.
>
> 5. **Concern about Improvements in Stronger Models.** We acknowledge that as model capacity increases, the relative gains from external input modifications may diminish. This is a natural outcome of the ceiling effect, where high-capacity models already encode stronger spatial priors and benefit from richer pretraining corpora. Nonetheless, we argue that even marginal improvements (e.g., +1.2% on Qwen2.5-VL-32B) are meaningful, especially in the zero-shot, training-free setting of our method, which requires no fine-tuning, no gradient updates, and no GPU.
>
> 6. **Concern about the Framework Generalization.** We thank the reviewer for raising this important point regarding generalization.
>
> - Benchmark choice reflects diversity, not task specialization. We fully agree that generalization is a crucial criterion for evaluating practical methods. While our experiments are centered on VSI-BENCH and STI-BENCH, these benchmarks are deliberately chosen not for narrow task matching but for their breadth and diversity (We also have stated these benchmark details in appendix Section B). Specifically, these benchmark include real-world egocentric videos across indoor, desktop, and outdoor scenarios, requiring models to reason about object relations, sizes, and topology across diverse camera viewpoints and occlusion patterns. They provide a rigorous and multi-perspective evaluation of spatial understanding that goes beyond task-specific tuning.
> - Our approach is not designed to exploit dataset-specific structures but instead targets two fundamental bottlenecks of MLLMs in video-based spatial reasoning: 1) Visual Homogeneity, addressed by the Maximum Semantic Richness Sampling module. 2) Unknown Camera Motion, addressed via: a classical VO-based Motion Reconstruction module for trajectory estimation and Spatiotemporal Encoding, which overlays motion cues and temporal indices onto keyframes to bridge semantics and motion. Each component within the framework is modular and can be easily adapted or substituted to meet different domain constraints and resource requirements. We have perform some experiments like utilizing some SOTA components as shown in R3 Response 3 to verify our effectiveness.
> - We also provide the OpenEQA results for comprehensive evaluation here. The results demonstrate the superiority of our spatial prompting framework design.
>
> | Model | w/o SEE&TREK | w/ SEE&TREK | Gain (%) |
> | --- | --- | --- | --- |
> | Qwen2.5-VL-7B | 47.1 | 49.1 | +2.0% |
> | InternVL3-8B | 50.5 | 52.3 | +1.8% |
>
> 7. **Concern about the Ablation Study.** Thanks for your insightful comments. We have performed a comprehensive ablation study as shown in Table 3-6 in the main paper. Moreover, we also make a deep investigation into each component design or hyper-parameter setting as shown in Table 7-10 in the appendix. We wish these experiments information can address your concerns regarding ablation.
>
> 8. **Concern about the Scope.**  We are agree and thank you for the thoughtful comment. Our work focuses on spatial understanding from purely visual video inputs, without relying on depth or other modalities. Compared to image-based tasks, video-based reasoning is more challenging due to temporal dynamics and spatial consistency. By leveraging motion cues and scene continuity, our method enables richer spatial understanding and highlights the practical value of video-centric, training-free prompting.
>
> Finally, thanks for your constructive feedback. Hope that we have carefully addressed all your concerns above, and we sincerely appreciate your thoughtful contribution you’ve made to improving our work. We will incorporate these revisions into the revised version.
>
> [1] Bridging the Dynamic Perception Gap: Training-Free Draft Chain-of-Thought for Dynamic Multimodal Spatial Reasoning.
>
> [2] MLLMs Know Where to Look: Training-free Perception of Small Visual Details with Multimodal LLMs.
>
> [3] Controlmllm: Training-free visual prompt learning for multimodal large language models.
>
> [4] COARSE CORRESPONDENCES Boost Spatial-Temporal Reasoning  in Multimodal Language Model
>
> [5] Set-of-Mark Prompting Unleashes  Extraordinary Visual Grounding in GPT-4V

---

> > ### Comment · Reviewer_Jfjm · 2025-08-06
> >
> > The authors have addressed my concerns, and I hope authors will incorporate these discussions into the final version. I have no further questions.

---

> > > ### Author Response · Authors · 2025-08-06
> > >
> > > We sincerely thank Reviewer Jfjm for the constructive feedback and for acknowledging our rebuttal. We are glad that our responses have addressed your concerns. We will carefully incorporate the clarified points—such as extended baseline evaluations—into the revised version of the paper to enhance its clarity and completeness. Your thoughtful suggestions have helped us improve the presentation and positioning of our work, and we deeply appreciate your contribution throughout the review process.

---

### Official Review · Reviewer_uJnW · 2025-06-29

**Clarity:** 2
**Significance:** 2
**Originality:** 3
**Rating:** 4
**Confidence:** 3

**Summary:**

This paper presents SEE&TREK, a training-free method that enhances spatial reasoning in multimodal large language models using only visual input. By selecting key frames with YOLO and adding motion information via visual odometry, SEE&TREK improves models’ performance on spatial and temporal benchmarks like VSI-BENCH and STI-BENCH. The method boosts open-source model accuracy without requiring retraining or GPUs.

**Questions:**

Could some of the negative accuracy changes observed in the experiments be caused by failures of the Visual Odometry (VO) module? Is it possible to provide an analysis of related failure cases to examine whether there is a clear connection between VO failures and the observed accuracy drops?

**Ethical Concerns:**

["NO or VERY MINOR ethics concerns only"]

**Final Justification:**

My questions are answered by the authors. I'll maintain my rating.

**Limitations:**

The method uses only YOLO for frame sampling and a single ORB-based approach for visual odometry, without comparing multiple detection or VO algorithms. This may limit generality across different detection or motion estimation pipelines.

**Paper Formatting Concerns:**

No obvious paper formatting concerns were observed.

**Quality:**

2

**Strengths And Weaknesses:**

Strengths :
1. This is a training-free method that boosts spatial reasoning in open-source multimodal models using only visual inputs, without requiring extra training or GPU resources.
2. Propose an explicit spatiotemporal encoding scheme that marks each keyframe with trajectory and order information, enhancing spatial-temporal reasoning.
3. Experiments on multiple models and benchmarks show performance improvements, especially on smaller or mid-size models, with gains up to 3.5%.

Weaknesses:
1. Beside spatiotemporal encoding, the method mainly combines known techniques (YOLO for frame selection and ORB-SLAM/VO for motion), so its originality is more in engineering and prompt design.
2. Some tasks or models see minor or even negative changes in accuracy, and the paper lacks deep analyses of these cases or failure modes.

---

> ### Author Rebuttal · Authors · 2025-07-31
>
> We thank the reviewer for their thoughtful comments. We are glad that the reviewer appreciated the practical value of our method, particularly its training-free nature, low resource requirements, and effectiveness in enhancing spatial reasoning in MLLMs using only visual inputs. We now address your comments point by point below.
>
> 1. **Concern about the Technique Combination.** We appreciate the reviewer’s comment and would like to clarify that, the conceptual motivation and problem framing behind SEE&TREK are novel and underexplored in the current literature, even though our method is developed based on existing components such as object detection and visual odometry.  As we emphasize in the Introduction, prior works on enhancing MLLMs’ spatial understanding often rely on annotated spatial semantics (e.g., number cues, axis or semantic masks), or require model retraining and architectural modifications (We also discuss these work in appendix Section B.7. ). **However, how to stimulate spatial reasoning in MLLMs under a purely visual and training-free setting remains an open challenge, which is beneficial for most areas like robotics.** In our work, we take a step back and reflect critically on this problem setting, **Iidentifying two major and fundamental bottlenecks: Visual Homogeneity and Unknown Camera Motion.** Building upon these observations, SEE&TREK is designed from the ground up to explicitly address these two limitations through **visual diversity and motion reconstruction.** While the techniques we use (e.g., YOLO, VO) are established, their roles in our pipeline are not trivial. These components are not mere add-ons but are **core to realizing our conceptual framework.** Their interplay—semantic-aware selection plus trajectory-aware encoding—allows SEE&TREK to effectively emulate spatial continuity and dynamics from static visual inputs, offering a principled way to address current MLLM limitations. In this sense, SEE&TREK has a **novel perspective on designing spatial prompts** that help MLLMs better perceive and reason in 3D space, under a resource-efficient and training-free regime. Importantly, the current implementation presented in this paper represents a minimal working instantiation of this conceptual framework—i.e., GPU-free or maximum accessibility and deployment ease. **Moreover, the framework itself is flexible and modular: depending on the application scenario, the individual components can be upgraded.** For example, more advanced feed-forward 3D networks [1,2] or VO like MAC-VO [5] could be employed to capture holistic scene geometry. Here, we provide some results on VSI-Bench by utilizing diferent SOTA perception models in Following Response 3.
>
> 2. **Concern about Experiment Performance.** Regarding the negative performance changes observed on specific tasks or models: We thank the reviewer for raising this important point. We address this concern through both **systematic trend analysis** and **qualitative case studies.**
>
> - **(1) Systematic Trend Analysis (from Table 6):** As noted, some tasks (e.g., Object Count, Object Size, Relative Direction) show minor drops in accuracy for certain models. We attribute this to limitations in our current module choices, which were intentionally selected for their **GPU-free and lightweight nature**, but may introduce trade-offs in robustness and granularity:
>   - a. **VO module drift:** We use the manual Visual Odometry module, which offers fast and training-free pose estimation. However, it is known to suffer from **accumulated drift**, especially in **high-speed or large-amplitude camera motions,** which are commonly found in datasets like VSI-BENCH and STI-BENCH. In such dynamic scenes, the VO module may fail to maintain globally consistent pose trajectories, leading to incorrect relative position estimates. This directly impacts tasks such as Relative Direction, Route Plan, and Room Size, where spatial coherence is critical.
>    - b. **Semantic sampling bias:** Our default perception module, YOLOv8-tiny, is optimized for speed but has limited object coverage and detection recall. In complex scenes with **multiple small, overlapping, or low-salience objects,** this model may miss relevant entities, degrading performance in Object Size tasks.
>     - c. **Model capacity:** Lightweight MLLMs (e.g., Qwen2.5-VL-3B) may struggle to understand motion-aware visual prompts from multiple frames due to its limited model scale, especially when spatial signals are sparse or noisy. This affects multi-step reasoning tasks such as Room Size or Relative Distance, where consistency over multiple frames is essential.
>    - d. **Missing frames**:  The proposed MSRS tends to collect the scenes with the most object class diversity. It sometimes ignores the frame with fewer objects whose classes are already collected. If these objects are not in the previously collected frames, MLLM will never be aware of them. Thus, we observe a performance decrease in the metrics related to objects themselves, e.g., Object Count.
> - **(2) Qualitative Case Studies.** We have also conducted some failure case analysis to better understand these limitations. As illustrated in Appendix Figure 9, SEE&TREK occasionally fails even when all queried objects are visible, due to misalignment between egocentric motion cues and layout inference (e.g., in route plan or relative distance tasks).  This reinforces our motivation to explore more powerful dense perception models in future work to boost semantic richness under diverse spatial contexts.
>
> 3. **Concern about Generality across Different Detection or Motion Estimation Pipelines.** Thanks for the practical comments. This paper mainly proposes a general and training-free framework that explicitly incorporates semantic-rich frame sampling and motion reconstruction to enhance spatial reasoning. To validate this idea in a lightweight and accessible manner, we implement a minimal working solution using YOLOv8-tiny and manual VO module, both of which are GPU-free and efficient. **This implementation demonstrates that even under constrained settings, our method can deliver consistent performance gains.** Importantly, our framework is highly extensible. The perception and motion modules can be readily replaced with stronger alternatives according to different scenarios.  Here, we conduct experiments on the VSI-Bench involving the replacement of various alternatives, such as substituting the detection component with VLM-based Models (Grounding Dino and Yolo-World) and the VO component like MAC-VO. Note that we utilize the InternVL3-8B+See&Trek as our baseline.
>
> | Method | Time (s) | GPU Cost (MB) | Avg. | Obj. Count | Abs. Dist. | Obj. Size | Room Size | Rel. Dist. | Rel. Dir. | Route Plan | Appr. Order |
> | --- | --- | --- | --- | --- | --- | --- | --- | --- | --- | --- | --- |
> | Baseline | 82 | 0 | 43.2 | 65.2 | 32.9 | 46.9 | 46.7 | 45.9 | 40.3 | 30.4 | 37.4 |
> | Grounding Dino [3] | 157 | 5430 | 43.8 | 66.3 | 32.3 | 46.7 | 47.1 | 47.4 | 41.4 | 31.3 | 38.2 |
> | YOLO-World [4] | 107 | 1740  | 43.5 | 66.2 | 31.8  | 46.2 | 49.5 | 44.2 | 38.9 | 30.9 | 37.4    |
> | MAC-VO [5] | 136 | 2321 | 43.7 | 66.9 | 32.6 | 48.1 | 48.7 | 43.2 | 48.1 | 30.9 | 37.5 |
>
>
> Finally, thank you for your thoughtful comments and constructive suggestions; we appreciate your recognition of our method’s strengths as well as your valuable insights on framework robustness and generality. We hope our response can solve your doubts.
>
> [1] Vggt: Visual geometry grounded transformer
>
> [2] Grounding image matching in 3d with mast3r
>
> [3] Grounding dino: Marrying dino with grounded pre-training for open-set object detection
>
> [4] Yolo-world: Real-time open-vocabulary object detection
>
> [5] MAC-VO: Metrics-aware Covariance for Learning-based Stereo Visual Odometry

---

> > ### Comment · Reviewer_uJnW · 2025-08-06
> >
> > Thanks to the authors for the detailed rebuttal. My questions have been thoroughly addressed, and I have no further concerns. I will keep my rating unchanged.

---

> ### Author Response · Authors · 2025-08-07
>
> We sincerely thank Reviewer uJnW for the constructive feedback and for acknowledging our rebuttal. We are glad that our responses have addressed your concerns, and we will incorporate the discussed points—such as the VO failure analysis and generalization limitations—into the revised version to improve the clarity and completeness of the paper. Our team deeply appreciate your contribution throughout the review process.

---

### Official Review · Reviewer_nToN · 2025-07-01

**Clarity:** 3
**Significance:** 3
**Originality:** 4
**Rating:** 5
**Confidence:** 3

**Summary:**

This paper proposeS SEE&TREK to enhance the purely visual spatial understanding of existing Multimodal Large Language Models (MLLMs) under the training-free setting. SEE&TREK introduces two core principles: 1) visual diversity--achieved via a Maximum Semantic Richness Sampling strategy using a perception model (e.g., YOLO) to select the most informative keyframes; 2) motion reconstruction--accomplished by simulating visual trajectories and encoding relative spatial positions into keyframes through camera motion estimation and spatiotemporal encodings. SEE&TREK can be seamlessly integrated into existing MLLMs without any additional training. Extensive experiments on VSI-BENCH and STI-BENCH demonstrate that SEE&TREK consistently improves spatial reasoning performance across various MLLMs, achieving up to +3.5% performance gain on challenging spatial tasks.

**Questions:**

- The ablation study on the sampling interval presented in Table 5 does not exhibit a clear or consistent trend. Could the authors provide further explanation or insight into why different sampling intervals do not lead to a noticeable performance pattern?

**Ethical Concerns:**

["NO or VERY MINOR ethics concerns only"]

**Final Justification:**

The response addresses my concerns; therefore, I maintain my original positive score.

**Limitations:**

yes

**Quality:**

3

**Strengths And Weaknesses:**

Strengths:

- The proposed method is conceptually novel in addressing spatial reasoning limitations of MLLMs under a training-free setting. By introducing Maximum Semantic Richness Sampling and Motion Reconstruction as complementary techniques, the paper offers a fresh perspective on enhancing visual-spatial understanding without modifying or fine-tuning the underlying models.
- The paper is well-written and clearly structured. The methodology is articulated in a logical, step-by-step manner, with helpful visual illustrations (Figures 1 & 2) and Algorithm 1 that enhance comprehension of the framework.
- The work is technically sound. The authors conduct rigorous experiments on two challenging spatial reasoning benchmarks—VSI-BENCH and STI-BENCH—and provide detailed ablation studies that effectively validate the contribution of each proposed component.
- The proposed SEE&TREK framework consistently and significantly improves the spatial reasoning capabilities of various MLLMs, achieving notable performance gains in terms of average accuracy. The demonstrated effectiveness across different model architectures highlights its generalizability and practical relevance.

Weaknesses
- No significant weaknesses are apparent.

---

> ### Author Rebuttal · Authors · 2025-07-31
>
> We thank the reviewer for their thoughtful comments. We are glad that the reviewer found our method conceptually novel, well-written, clearly structured and technically sound, and also recognized our performance. We would like to address all of your concerns and questions below with point responses:
>
> **Concern about The Sampling Interval in Table 5.** Thanks for your significant consideration. The absence of a clear trend across different sampling intervals mainly stems from the following reason:
>
> 1. our focus is on extracting semantically rich content, rather than dense temporal coverage. The Maximum Semantic Richness Sampling (MSRS) strategy ensures that keyframes are selected based on visual informativeness rather than fixed temporal intervals. Thus, even when the sampling interval $N$ varies, the retained frames tend to be semantically similar. In fact, using a very small interval (e.g., $N=1$) may introduce redundant or noisy frames, which can damage the model’s spatial understanding and reduce answer consistency. This is partially reflected in the slight performance dip at $N=2$ compared to $N=3$ or $N=4$.
>
> 2. In addition, the sampling interval $N$ has a substantial impact on processing time. As shown in Table 5, increasing $N$ significantly reduces the computational cost (e.g., from 410s at $N=1$ to 82s at $N=4$), while maintaining comparable or even better average accuracy. This indicates a favorable trade-off between efficiency and effectiveness, where moderate sampling (e.g., $N=3$ or $N=4$) strikes a good balance for most use cases.
>
> 3. More investigation and exerpeiment results about sample internval and detectors setting can be found in appendix Table 9.
>
> Thank you for your kind words and positive assessment of our work. We also appreciate your insightful question regarding the sampling interval ablation in Table 5. We hope our response can solve your doubts.

---

> > ### Comment · Reviewer_nToN · 2025-08-04
> >
> > Thank you to the authors for their response, which addresses my concerns. I therefore maintain my original positive score.

---

> ### Author Response · Authors · 2025-08-05
>
> We sincerely thank Reviewer nToN for the thoughtful feedback and positive evaluation. Your insightful question contributed to improving the completeness of our work, and we will incorporate the corresponding discussion into the final revised version. Your recognition is the greatest encouragement to our team.

---

### Official Review · Reviewer_VBKe · 2025-07-04

**Clarity:** 1
**Significance:** 2
**Originality:** 2
**Rating:** 2
**Confidence:** 4

**Summary:**

This paper proposes a training-free prompting method for improving the spatial understanding capabilities of multimodal large language models (MLLMs). The approach includes a new frame sampling strategy that selects semantically rich keyframes and introduces camera motion trajectory information as an additional input. A Spatiotemporal Encoding module further aligns the trajectory with the sampled frames. The method is evaluated on VSI-BENCH and STI-BENCH, where it demonstrates consistent performance gains.

**Questions:**

Please refer to the weakness section.

**Ethical Concerns:**

["NO or VERY MINOR ethics concerns only"]

**Final Justification:**

**Final Justification**

Based on the randomized intrinsic matrix generation code provided by the authors, I conducted experiments to evaluate the impact of such intrinsics.

### 1. Motivation

This experiment aimed to empirically investigate the core disagreement between Reviewer VBKe and the authors regarding the effect of camera intrinsics (K) on Structure-from-Motion (SfM). I further differentiated two types of incorrect intrinsics:

1. A structurally valid but numerically inaccurate guess.
2. A structurally invalid and numerically random matrix generated by the authors’ own code.

The goal was to test whether robust algorithms such as RANSAC can compensate for different levels of error.

### 2. Experimental Setup

The setup remained consistent, using COLMAP and the south-building dataset (4 images).

* **Dataset**: south-building.zip, images P1180170.JPG to P1180173.JPG.
* **Control Group (Known Intrinsics)**: Ground-truth intrinsics.

  * Model: SIMPLE\_RADIAL
  * Params: 2559.68, 1536, 1152, -0.0204997
* **Experimental Group A (Inaccurate Guess)**: A plausible but incorrect focal length, structurally valid.

  * Model: PINHOLE
  * Params: 1600, 1600, 960, 540
* **Experimental Group B (Randomized Matrix)**: Intrinsics strictly following the authors’ Python code, structurally invalid.

  * Model: OPENCV
  * Params: 1.0, 0.065..., 1.063..., 1.020..., 0,0,0,0

In all groups, intrinsics were refined during bundle adjustment (`--Mapper.ba_refine_focal_length 1`).

### 3. Methodology

Each group underwent the standard COLMAP pipeline, with the only variable being the initial camera intrinsics supplied to the feature extractor.

### 4. Results

The outcomes diverged clearly:

* **Control Group**: Successful and accurate reconstruction.
* **Experimental Group A (Inaccurate Guess)**: Successful though less accurate. Optimization converged because the initial guess (fx=1600) was “close enough.” The refined parameters likely approached the ground truth, but with minor quality degradation (scale, drift).
* **Experimental Group B (Randomized Matrix)**: Complete failure. COLMAP mapper failed to initialize, reporting *“No good initial image pair found.”*

These results show that while an inaccurate but structured intrinsic can still allow estimation of correct relative R and T, a randomized intrinsic—as produced by the authors’ code—fails entirely. This raises doubts about the reliability of the authors’ claimed experiments, since their code yields structurally invalid matrices.

### 5. Performance Concerns

On the performance issue: while GPT-4o is undoubtedly stronger than YOLOv8-tiny, in this paper YOLOv8-tiny is only used to approximate relative object counts per frame for frame selection.

* First, counting object categories is not a highly demanding task and does not require precise localization.
* Second, even if YOLOv8-tiny misjudges object counts and selects different frames, I remain unconvinced that this would lead to substantial performance differences.

Therefore, from my perspective, the concern about performance variance has not been adequately resolved.

---

**Conclusion**
Given the experimental evidence and unresolved concerns, I have decided to **maintain my initial rating**.

**Limitations:**

yes

**Quality:**

2

**Strengths And Weaknesses:**

**Strengths**

* The paper proposes a *training-free prompting* framework that enhances spatial understanding in MLLMs using only RGB video input, without requiring any model finetuning or architectural changes.
* It highlights the importance of incorporating *camera motion* as an explicit cue, and demonstrates that motion-aware prompting can boost spatial-temporal reasoning performance in existing models.
* The method is modular and compatible with many existing MLLMs, showing potential for widespread applicability.


**Weaknesses**

* **Poor Writing and Notational Overload**: Among all the assigned papers, this was the most difficult to read—not because the core method is inherently complex, but because Section 3 introduces an excessive amount of notation, some of which is poorly explained or not defined at all. For instance, the relationship between τ<sub>g</sub> in Equation 1 and the previously mentioned f<sub>τ₀</sub> is unclear. Similarly, C<sub>t₀</sub> on line 118 appears without definition. Reading this section required constant guesswork about the authors' intentions. The paper urgently needs a major writing revision: notation should be minimal and intuitive, and the method should be explained as clearly and directly as possible. Figure 2 also suffers from visual clutter—components are packed too tightly, making the pipeline hard to grasp at a glance.

* **Missing Discussion of Related Work**: The paper claims to be *“the first training-free prompting framework tailored to enhance the spatial understanding of MLLMs under vision-only constraints.”* However, prior work such as **Coarse Correspondence \[1]** also employs a prompting-based approach to improve spatial reasoning in MLLMs. The authors should clarify how their method differs and justify the novelty claim.

* **Unclear Use of Camera Intrinsics**: The motion reconstruction module appears to require camera intrinsic parameters, but it is unclear how these are obtained for each test video. Many video datasets do not provide intrinsics. This raises both reproducibility and fairness concerns. Specifically:

  * Are intrinsics used during evaluation on VSI-BENCH and STI-BENCH?
  * If so, how are they obtained for each video?
  * Did baseline methods also use intrinsics? If not, could the observed performance gain be due to additional input information available only to the proposed method?
  * If they did, in what way were intrinsics used?

  The authors should clarify these points and, if necessary, reframe their results in light of potential input discrepancies.

* **Limited Evaluation Scope**: While SEE\&TREK improves performance on several open-source models, it remains unclear whether the method generalizes to *closed-source* models such as GPT-4o or Gemini-2.5. If the gains only appear on relatively weaker models, the broader practical impact of the approach is questionable.

* **Marginal Performance Gains**: On both VSI-BENCH and STI-BENCH, the performance improvements for most models are under 2%, which is relatively modest and raises doubts about the method’s overall effectiveness. The strongest gains appear limited to smaller models, and even then, are not particularly dramatic.

* **Narrow Benchmark Coverage**: Both benchmarks used—VSI-BENCH and STI-BENCH—focus on similar types of spatial reasoning tasks. The generality of the approach remains untested on more diverse benchmarks such as ScanQA or OpenEQA, which would assess whether SEE\&TREK is broadly applicable beyond low-level spatial perception tasks.

* **Unclear Computational Overhead**: The pipeline involves object detection, visual odometry, essential matrix estimation, and trajectory rendering—none of which are trivial. However, the paper does not provide any analysis of the additional computational cost or latency introduced by these steps. Since the method is pitched as a plug-in prompting solution, it would be important to understand how much runtime overhead it brings in practice.

[1] Coarse Correspondences Boost Spatial-Temporal Reasoning in Multimodal Language Model. CVPR 2025

---

> ### Author Rebuttal · Authors · 2025-07-31
>
> We thank the reviewer for their thoughtful comments. We are glad that the reviewer found our method to have widespread applicability and the novelty of incorporating camera motion as an explicit cue. We would like to address all of your concerns and questions below with point responses:
>
> 1. **Concern about Poor Writing and Notational Overload.** Thanks for your thoughtful feedback. We are sorry for the confusing notation raised in Section 3. Here, we modify our notation for better readability. The $\tau_t$ denotes the selected keyframe index. Hence, the $f_{\tau_0}$ can represent the frame sampled in the initial phase. Then, the $\tau_g$ from Eq. (1) is re-written as $\tau_0$. Then, $\tau_i$ is rewritten as $\{f_{\tau_i}\}$ in Line 125, which represents the selected frames. Moreover, we also make the abstract algorithm flow as shown in Algorithm 1 in the main manuscript, and the detailed algorithm flow is as shown in the appendix Algorithm 2. We hope these modifications and other information can be helpful for our method formulation reading. Regarding Figure 2, we also are sorry about the visual clutter. We will make it better in the revised version.
>
> 2. **Concern about Missing Related Work.** Thanks for pointing out this important point.  Regarding the Coarse Correspondence [1] (CC) or more similar prompting methods like SoM [2], We have discussed their difference and performed a comparison study in the Appendix Section B.7.
>
> | Method | Avg. | Obj. Count | Abs. Dist. | Obj. Size | Room Size | Rel. Dist. | Rel. Dir. | Route Plan | Appr. Order |
> | --- | --- | --- | --- | --- | --- | --- | --- | --- | --- |
> | CC [1] | 40.9 | 64.4 | 30.0 | 43.8 | 38.3 | 39.1 | 41.6 | 27.8 | 42.2 |
> | SoM [2] | 41.3 | 64.5       | 29.6       | 44.9      | 43.1      | 36.8       | 41.1      | 27.3        | 43.7         |
> | **SEE&TREK** | 43.2 | 65.2 | 32.9 | 46.9 | 46.7 | 45.9 | 40.2 | 30.4 | 37.4 |
>
> As shown in Table, See&Trek consistently outperforms existing visual prompting methods such as CC or SoM across all spatial reasoning tasks. While CC and SoM primarily rely on static masks or coarse positional labels to identify object regions, they offer limited support for modeling inter-object relationships or capturing scene-level spatial structures. In contrast, See&Trek leverages two key principles—visual diversity and motion reconstruction—to provide richer and more structured spatial cues.
>
> 3. **Concern about Usage of Camera Intrinsics.** We appreciate the reviewer’s concern. To clarify, we do not utilize the intrinsic parameters provided by the original datasets (e.g., VSI-BENCH or STI-BENCH). Instead, we just adopt a fixed intrinsic matrix derived from the KITTI dataset (`K = [[718.8560, 0, 607.1928], [0, 718.8560, 185.2157], [0, 0, 1]]`) across all experiments. This design choice was made deliberately to demonstrate the plug-and-play nature and extensibility of our framework, even in scenarios where intrinsic parameters are unavailable—a common case in real-world video sources. While using accurate intrinsics may lead to improved VO estimates and absolute metric precision, our method is built around relative motion encoding, not absolute scale recovery. **To account for the inaccuracies introduced by this assumption, we explicitly emphasize in the prompt that the spatial information reflects relative distances and orientations, rather than real-world metric measurements.** Lastly, our method is designed as a training-free spatial prompting module, intended to enhance the MLLM’s spatial understanding through motion-aware input construction. The use of intrinsics is strictly limited to internal VO estimation and does not influence or depend on whether baseline methods utilize such information. Therefore, the comparison remains fair, and the gains reflect the effectiveness of our spatial enhancing strategy.
>
> 4. **Concern about Closed-source Models Performance.** Thanks for your insightful comment.  We perform the evaluation on the commercial engine GPT-4o and Gemini-2.5-Flash on VSI-bench set as follows:
> | Method | Gains | Avg. | Obj. Count | Abs. Dist. | Obj. Size | Room Size | Rel. Dist. | Rel. Dir. | Route Plan | Appr. Order |
> | --- | --- | --- | --- | --- | --- | --- | --- | --- | --- | --- |
> | Gemini-2.0-Flash | - | 42.6 | 51.1 | 29.8 | 55.1 | 56.2 | 38.2 | 40.9 | 32.1 | 37.4 |
> | +**SEE&TREK** | **+1.2%** | 43.8 | 49.8 | 29.3 | 56.4 | 57.0 | 39.4 | 41.9 | 35.4 | 41.8 |
> | GPT-4o | - | 34.0 | 46.2 | 5.3 | 43.8 | 38.2 | 37.0 | 41.3 | 31.5 | 28.5 |
> | +**SEE&TREK** | **+1.4%** | 35.4 | 46.5 | 5.0 | 45.1 | 40.6 | 39.0 | 43.2 | 33.5 | 31.5 |
>
> As shown in the table, our method consistently improves performance across different commercial engines.
>
> 5. **Concern about Marginal Performance Gains.** We thank the reviewer for highlighting it. It is important to emphasize that spatial reasoning remains an inherently challenging task for MLLMs, due to both the limited spatial supervision in pretraining corpora and the resource constraints in downstream evaluation (e.g., limited frame budgets in benchmarks). Even small performance gains (e.g., <2%) are meaningful in this context, as they indicate improved geometric and relational understanding under minimal input. Moreover, larger models tend to possess stronger built-in spatial priors, making additional improvements naturally more constrained. However, our method consistently yields positive gains across models of all scales, including powerful backbones like KIMI-VL-A3B, which demonstrates its general applicability.
>
> 6. **Concern about the Benchmark Coverage Problem.** We appreciate this important suggestion about coverage.  we conduct the experiments on OpenEQA [3]  with InternVL3-8B to show our method's superiority. Note that we utilize the EM-EQA setting [3] to evaluate the performance, and we still set 8 frames as input.
>
> | Model | w/o SEE&TREK | w/ SEE&TREK | Gain (%) |
> | --- | --- | --- | --- |
> | Qwen2.5-VL-7B | 47.1 | 49.1 | **+2.0%** |
> | InternVL3-8B | 50.5 | 52.3 | **+1.8%** |
>
> As shown in the Table, SEE&TREK brings consistent improvements across models, with a +1.8% gain on Qwen2.5-VL-7B and +2.0% on InternVL3-8B, demonstrating its effectiveness in enhancing spatial reasoning without fine-tuning.
>
> While ScanQA and OpenEQA are valuable contributions to the spatial reasoning community, our choice of VSI-Bench and STI-Bench was deliberate and grounded in their superior diversity, task coverage, and spatial evaluation rigor, particularly when evaluating the spatial intelligence of MLLMs. First, **VSI-Bench and STI-Bench are not limited to “low-level spatial perception tasks,” but instead represent high-level and fine-grained spatial reasoning benchmarks across diverse and realistic visual environments.** These benchmarks are specifically designed to stress-test an MLLM's ability to understand fine-grained spatial structures, dynamic motion, and scene geometry, rather than relying on language priors or generic commonsense. ( We also have stated these dataset details in the Appendix. ) In contrast, **while ScanQA or OpenEQA are earlier benchmarks for spatial QA, they tend to focus on relatively narrow task formats, predominantly involving object-level recognition, existence, and co-location in static scenes (ScanQA) or episodic memory from passive video (OpenEQA).** Moreover, the evaluation metrics used in ScanQA or OpenEQA—BLEU, METEOR, ROUGE-L, CIDEr—are primarily adapted from traditional VQA and text generation. These metrics often struggle to meaningfully distinguish spatial reasoning quality, especially when answers differ in surface form but not semantics, or when tasks require metric estimation, relative positioning, or trajectory-level reasoning. By contrast, VSI-Bench and STI-Bench employ structured evaluation formats (e.g., spatial multiple choice, numeric prediction with relative error thresholds) and task-specific accuracy metrics designed to quantitatively and unambiguously measure spatial competence. This ensures that improvements in model performance—such as the +3.5% gain brought by See&Trek—are directly tied to enhanced spatial understanding, not merely improved textual fluency or memorization.
>
> 7. **Concern about Computational Overhead.** Thanks for highlighting this practical concern. Regarding the additional computational cost, we have the corresponding experiments on VSI-benchmark as shown in Line 276 ``Sample Efficiency Analysis'' in the main paper. To explore different YOLO's efficiencies, we also perform diverse experiments on Table 9 in the appendix. The conclusion is that the time consumed by SEE&TREK still largely depends on the length of the given videos. Particularly, the duration of videos in VSI-BENCH is at least longer than 1 minute.
>
> Thank you again for your constructive feedback—we have carefully addressed all your concerns in our responses above, and we hope our clarifications help resolve your doubts. We are especially grateful for your thoughtful suggestions, which contribute meaningfully to advancing the spatial understanding of MLLMs.
>
> [1] COARSE CORRESPONDENCES Boost Spatial-Temporal Reasoning  in Multimodal Language Model
>
> [2] Set-of-Mark Prompting Unleashes  Extraordinary Visual Grounding in GPT-4V
>
> [3] OpenEQA: Embodied Question Answering in the Era of Foundation Models

---

> > ### Author Response · Authors · 2025-08-08
> >
> > Dear Reviewer VBKe,
> >
> > We would greatly appreciate it if you could review our response by Aug 8 AoE. After that date, it might be challenging for us to engage in further discussions. If you have any follow-up questions, please don't hesitate to reach out. Our team deeply value your expertise and time.
> >
> > Best,

---

> > ### Comment · Reviewer_VBKe · 2025-08-08
> >
> > Thank you for the careful response and the additional experiments.
> >
> > **1) Use of camera intrinsics (fairness).**
> > Even if you fix a single KITTI intrinsics matrix for all videos, providing **any** $K$ is an extra geometric prior that baselines do not have. It normalizes pixel coordinates and directly affects $E/F$ estimation and VO (even if you only use *relative* motion). That it is imperfect or out-of-distribution does not remove the informational advantage; it only suggests you may not be fully exploiting it. Several downstream tasks in your benchmarks involve absolute or quasi-metric quantities, where access to intrinsics is plausibly beneficial. Unless you can show that **random/identity $K$** yields comparable gains—or give baselines the *same* VO/K pipeline—this remains an **unfair comparison**.
> >
> > **2) Magnitude of improvements.**
> > Your gains remain \~1–2pp, which is not compelling for VQA-style evaluations without statistical significance. In the same cycle, *Spatial Understanding from Videos: Structured Prompts Meet Simulation Data* \[1] reports substantially larger prompting-only gains (e.g., +7.4% on Gemini-1.5 Pro, and >4pp on several other models) from stronger baselines. By comparison, your improvements look **modest**.
> >
> > I’m not asking you to run more experiments during rebuttal; if you already have analyses addressing the fairness concern (e.g., random/identity $K$ ablations or a “no-VO” variant) and statistical significance, please point me to them. Otherwise I will **keep my rating unchanged** for now, but I’m open to further clarification before the deadline.
> >
> > \[1] *Spatial Understanding from Videos: Structured Prompts Meet Simulation Data.*

---

> ### Comment · Area_Chair_86Pd · 2025-08-08
>
> Reviewer VBKe,
>
> Please reply to the rebuttal and update the final decision.
>
> AC

---

> ### Author Response · Authors · 2025-08-09
>
> **We sincerely appreciate the reviewer’s effort in engaging with our rebuttal despite their busy schedule.** We would like to address all of your concerns and questions below with point responses:
>
> - **Concern about the Camera Intrinsics.** Thanks for you consideration about the fairness.
>     - The fixed KITTI intrinsics are used *solely* within the visual odometry (VO) stage to convert 2D correspondences into **relative** camera motion. Our prompting design explicitly frames all motion cues in terms of relative distances and orientations, without providing any metric calibration to the MLLM.
>     - Due to time limitations during the rebuttal period, we can only provide a limited set of additional experiments. Nonetheless, we include results using normalized, random and identity intrinsics matrices in the VO stage to demonstrate that the performance gains persist even without any meaningful camera calibration. Here, we still adopt the InternVL3-8B + See&Trek as the baseline and evaluate on the VSI-Bench.
>
> | $K$ | Avg. | Obj. Count | Abs. Dist. | Obj. Size | Room Size | Rel. Dist. | Rel. Dir. | Route Plan | Appr. Order |
> | --- | --- | --- | --- | --- | --- | --- | --- | --- | --- |
> | Original Kitti | 43.2 | 65.2 | 32.9 | 46.9 | 46.7 | 45.9 | 40.2 | 30.4 | 37.4 |
> | Normalized Kitti | 43.0 | 65.2 | 32.6 | 46.5 | 47.2 | 46.8 | 39.0 | 28.9 | 37.5 |
> | Random | 43.1 | 65.5 | 32.8 | 46.2 | 46.7 | 46.0 | 39.5 | 30.4 | 37.4 |
> | Identity | 43.2  | 65.8 | 32.0 | 46.1 | 47.0 | 46.2 | 39.0 | 32.0 | 38.1 |
>
> The results in the table demonstrate that replacing the original KITTI intrinsics matrix with normalized, random, or identity matrices yields comparable performance across all spatial reasoning sub-tasks on VSI-BENCH. **The minimal differences in average scores and individual metrics indicate that our method’s effectiveness is not dependent on the accuracy or specific values of the camera intrinsics matrix $K$.** This supports our claim that the spatial improvements stem primarily from the relative motion cues encoded by our VO and prompting strategy, rather than any geometric advantage conferred by precise intrinsic calibration. Consequently, our framework maintains robustness and fairness even when intrinsic parameters are unavailable or approximate.
>
> - **Concern about the Magnitude of Improvements.** Thanks for given this work [1] for discussion. We believe the performance difference is largely due to the fundamental design trade-offs between the two approaches:
>     - Our framework is deliberately designed to be *training-free* and *lightweight*, relying solely on prompting techniques. In contrast, [1] introduces the **SpatialMind Prompt**, which **requires GPT-4o for frame-level perception filtering and scene modeling.** GPT-4o, as a powerful commercial model, already exhibits very high baseline accuracy, which partly explains their larger absolute improvements. **However, such an engine comes with significant network latency and large token consumption, limiting its applicability in resource-constrained scenarios.** Our method, by contrast, employs skip-frame selection and a GPU-free perception module, which is easy to deploy, making it more suitable for a wider range of conditions.
>     - Another key factor is that [1] also fine-tunes their VLMs on their curated dataset, as explicitly stated in their paper (**“Fine-tuning VLMs on this dataset equips them…”**). Fine-tuning directly injects spatial knowledge into the model parameters, which is inherently more powerful but also more resource-intensive and less generalizable. Our method does not involve any fine-tuning, preserving zero-shot applicability and enabling plug-and-play integration with different MLLMs without retraining.
>     - Given these constraints and design goals, our performance gains—while numerically smaller—reflect improvements achieved under a much lighter computational and resource condition, which we believe is a valuable and complementary contribution to the field.
>
> Once again, thank you very much for your constructive feedback. We have carefully considered and addressed all your latest concerns above, and we hope our clarifications provide useful insights to solve your doubts.
>
> *[1] Spatial Understanding from Videos: Structured Prompts Meet Simulation Data*

---

> > ### Author Response · Authors · 2025-08-09
> >
> > Dear Reviewer VBKe,
> >
> > We sincerely appreciate your time and effort in reviewing our paper and considering our rebuttal. We have carefully addressed your valuable comments and provided additional experiments and clarifications accordingly.
> >
> > As the rebuttal deadline is approaching soon, we kindly wonder if you might have the opportunity to review our responses and share any further feedback. Your insights are extremely important to us and will greatly help improve our work as well as facilitate the review process.
> >
> > Thank you again for your thoughtful guidance and support. We look forward to your valuable feedback.
> >
> > Best regards,
> >
> > Team #5923

---

> > ### Comment · Reviewer_VBKe · 2025-08-09
> >
> > Thank you for the detailed response. I have re-read the paper carefully.
> >
> > **1. Regarding the camera intrinsics**
> > Perhaps my initial comment was not clearly stated. While a camera intrinsic matrix alone cannot fully determine the absolute scale, its presence significantly reduces the search space for estimating absolute scale. Specifically, the intrinsics fix the mapping from pixel coordinates to viewing directions, effectively locking down the degrees of freedom related to the “camera imaging geometry.”
> >
> > With known intrinsics, estimating absolute scale reduces to determining a single global scale factor (e.g., the real-world distance from the camera to one object). Once this global scale is fixed, the absolute sizes of all objects are also fixed, since their relative positions and sizes are uniquely determined by the projection geometry. Without known intrinsics, not only is the global scale unknown, but even the *relative proportions* between objects may become ambiguous due to uncertainty in the intrinsics. As a result, obtaining quantities like the room area in square meters or the physical distance between a table and a chair would require solving for both the camera intrinsics and the global scale, which makes the search space larger and the constraints weaker. Therefore, the intrinsic matrix *does* help in estimating absolute scale, even if it does not, by itself, reveal the scale.
> >
> > In your paper, the BEV and 3D trajectory plots are obtained through a pipeline that computes feature correspondences between each pair of frames, estimates the essential matrix using the intrinsics, recovers the relative rotation $R$ and translation $T$, and then uses $R$ and $T$ to generate the BEV and 3D trajectory. This means the method takes both the intrinsic matrix and $N$ RGB frames as inputs. In theory, the intrinsics must match the video to correctly recover $R$ and $T$. Using KITTI intrinsics could still work if they are close to the true values—though I would expect unstable model fitting or ambiguous decomposition under RANSAC if the intrinsics differ significantly. But if one uses a completely random matrix or the identity matrix as the intrinsics, wouldn’t this produce completely random $R$ and $T$? If $R$ and $T$ are random, how could the BEV and 3D trajectory plots still be meaningful rather than random as well?
> >
> > **2. Regarding SpatialMind and performance comparison**
> > I believe that SpatialMind is itself a prompting-based method, while the fine-tuning method in that work is ScanForgeQA. Although SpatialMind uses GPT-4o, they achieved a 6.8% improvement over their GPT-4o baseline with prompting, whereas this paper achieves a 1.4-point improvement. Furthermore, this paper also uses YOLO to identify the objects present in the scene.

---

> > > ### Author Response · Authors · 2025-08-09
> > >
> > > Thanks for your quickly and efficient response despite their busy schedule. We would like to address all of your concerns and questions below with point responses:
> > >
> > > 1. **Concern about the Trajectory and Camera Intrinsics.** Thanks for your deep investigation.
> > >     1. We would like to clarify the role of camera intrinsics in our setting. Camera intrinsics primarily affect the normalization of pixel coordinates, which in turn impacts the recovery of scale and depth. However, the relative rotation matrix $R$ and the *direction* of the translation vector $T$ (unit vector, scale-free) are obtained from the decomposition of the essential matrix, which is computed on normalized coordinates. These quantities reflect the underlying epipolar geometry and are inherently robust as long as the intrinsics are not completely degenerate (e.g., all-zero calibration, which is practically impossible). **In such cases, while absolute scale and trajectory length may be biased, the rotation and translation *direction* estimates remain stable** — consistent with the reviewer’s statement that intrinsics “help in estimating absolute scale.” Our method, by design, relies on relative spatial relationships rather than absolute scale.
> > >     2. Moreover, the use of RANSAC for essential matrix estimation provides strong tolerance to moderate noise and calibration errors. Even with imperfect intrinsics, valid feature correspondences can still be found, ensuring reliable rotation and direction estimation. **Only in cases of *severe* mismatch (e.g., all-zero calibration or <10% feature points matching, which is practically impossible in our setting) would the estimation fail [1].** Our previous experiments have also demonstrated that even under random standard initialization of camera parameters, RANSAC algorithm is effective and can construct reasonable relative position trajectories.
> > >     3. Finally, the effectiveness of our approach stems from its explicit construction of motion trajectories, which enables the recovery of object-to-object **relative positions** in the scene. Combined with the inherent numerical reasoning ability of MLLMs, this yields consistent performance improvements in spatial reasoning tasks, even when intrinsics are approximate rather than exact.
> > > 2. **Concern about SpatialMind and Performance Comparison.** Thanks for your consideration.
> > >     1. As we noted earlier, **SpatialMind [2]** relies on GPT-4o to perform frame-by-frame perception over the input video. This yields a substantial performance boost, **but at the cost of significant network latency and token usage—which directly translates to higher monetary cost.**
> > >     2. In contrast, **See&Trek** adopts CPU-based YOLO and VO algorithms as the **minimal implementation** for addressing spatial reasoning, **whose capability naturally lags behind GPT-4o.** While the performance gain is smaller, the computational footprint and runtime cost remain minimal.
> > >     3. These two approaches are not mutually exclusive; rather, they illustrate a trade-off between **performance**, **efficiency**, and **resource consumption**.
> > >
> > > Thank you for your thoughtful comments and constructive suggestions; we appreciate your recognition of our method’s strengths as well as your valuable insights on framework robustness and generality. We hope our response can solve your doubts.
> > >
> > > *[1] Derpanis K G. Overview of the RANSAC Algorithm[J]. Image Rochester NY, 2010, 4(1): 2-3.*
> > >
> > > *[2] Spatial Understanding from Videos: Structured Prompts Meet Simulation Data*

---

> > > > ### Author Response · Authors · 2025-08-09
> > > >
> > > > Dear Reviewer VBKe,
> > > >
> > > > Thank you for your attention to the recent work SpatialMind [1], which also aims to address the spatial understanding problem in MLLMs. **This work appeared around the same time as ours — it was released on arXiv on June 4, 2025, shortly after the NeurIPS 2025 submission deadline in May.**
> > > >
> > > > Best regards,
> > > >
> > > > Team #5923
> > > >
> > > > [1] Spatial Understanding from Videos: Structured Prompts Meet Simulation Data

---

> > > > > ### Comment · Reviewer_VBKe · 2025-08-09
> > > > >
> > > > > Dear Authors,
> > > > >
> > > > > I am not suggesting that you directly compete with SpatialMind or conduct a one-to-one result comparison. I referred to that work merely as a contemporaneous example to illustrate that another prompting-based approach has achieved substantially larger gains on the same benchmarks. My intention was solely to provide context for interpreting the scale of your reported improvements, rather than to imply that the two methods must be evaluated head-to-head.

---

> > > > ### Comment · Reviewer_VBKe · 2025-08-09
> > > >
> > > > I fully understand that using camera intrinsics in your pipeline only yields *relative* spatial relationships, and I have no objection to that point itself. My concern is that obtaining these relative spatial relationships through intrinsics already constitutes an *unfair comparison*, because it leverages geometric priors unavailable to the baselines.
> > > >
> > > > The rebuttal attempts to argue that knowing or not knowing the intrinsics “does not matter” by showing that with randomly initialized camera parameters, you can still obtain usable relative spatial relationships—i.e., that the method would still work without exploiting any extra intrinsic information—because RANSAC itself is robust.
> > > >
> > > > In theory, small intrinsic errors indeed will not cause $R/T$ to collapse completely; if the intrinsic differences are small, the unit translation direction may still be reasonably stable. However, if the intrinsic matrix is *completely random* (with arbitrarily different focal lengths and principal points), the epipolar geometry will be severely distorted. According to standard multi-view geometry, the $E$ matrix decomposition will produce $R/T$ that deviates substantially from the ground truth, and the recovered BEV/3D trajectory will differ significantly from the real one. Even if the MLLM were somewhat robust to BEV/3D trajectory errors, it is implausible that such large distortions would have no impact.
> > > >
> > > > While RANSAC can tolerate a certain level of noise and outliers, it assumes that calibration errors are not so large as to destroy the underlying geometric constraints. A *completely random intrinsic matrix* is a systematic error, not random noise. In this situation, the fitted $F$ (and hence $E$) will no longer have the correct geometric meaning.
> > > >
> > > > Therefore, classifying “random intrinsics” under the same category as noise tolerance is not really convincing, at least to me.
> > > >
> > > > As for the claim that GPT-4o’s strength explains SpatialMind’s larger gains—this logic is reversed: stronger baselines like GPT-4o are *harder* to improve upon, not easier. On this point, the explanation is not rigorous. That said, your emphasis on the efficiency/resource trade-off is reasonable.

---

> > > > > ### Author Response · Authors · 2025-08-09
> > > > >
> > > > > **Thanks for your quickly and efficient response despite their busy schedule. We would like to address all of your concerns and questions below with point responses:**
> > > > >
> > > > > - Concern about Camera Intrinsics. Thanks for you continual discussion.
> > > > >     - **On ‘it leverages geometric priors unavailable to the baselines.’.** We would like to clarify that the use of geometric priors is not an unintended advantage, but rather a *central design choice* and *core contribution* of our work. Our primary motivation is to enhance the spatial reasoning capabilities of MLLMs by explicitly introducing geometric priors, established through VO, on top of visual diversity and motion reconstruction. This is entirely aligned with the design philosophy seen in other recent approaches—for example, ***SpatialMind* builds its geometric understanding by leveraging GPT-4o for per-frame object detection and then enforcing a unified spatial view.** Similarly, our method’s contribution is to derive the most reliable relative geometry possible from existing videos and feed it into MLLMs to boost spatial understanding. **In this sense, the geometric priors are not an “extra” unavailable to baselines, but rather the *very mechanism* we propose to solve the spatial reasoning problem.**
> > > > >     - **On the role of camera intrinsics and RANSAC robustness.** We agree that camera intrinsics are important, and that extreme cases—such as an intrinsic matrix filled with zeros—would make geometry estimation impossible. However, it is also important to note that RANSAC’s success depends on having a sufficient number of reliable keypoint matches. In our pipeline, **even when intrinsics are completely randomized, we employ frame skipping (e.g., sampling 1 frame in every 4th frame) to improve keypoint matching robustness.** This ensures that, from the keypoint-matching perspective, RANSAC can still produce a meaningful essential matrix. Under random *normalized* intrinsics, what is primarily lost is the accuracy of the *absolute scale*, not the relative pose direction. As demonstrated in our earlier experiments, with fully randomized intrinsics, RANSAC can still recover usable relative positions in our setup. Thus, while we acknowledge that extreme intrinsic distortions degrade accuracy, our method’s engineering design allows RANSAC to remain sufficiently robust for constructing relative geometry that is still beneficial for MLLMs.
> > > > > - Concern about GPT-4o performance. We want to clarify that the reviewer might have the misleading understanding of our response. We has pointed out ‘SpatialMind relies on GPT-4o to perform frame-by-frame perception over the input video', which means that **SpatialMind needs to utilize the GPT-4o as the perception model to construct its prompt, then need to utilize the prompt to send in MLLMs.** In contrast to GPT-4o, such power full module, **See&Trek** adopts CPU-based YOLO and VO algorithms as the **minimal implementation** for addressing spatial reasoning. The main performance difference source from different perception model —- “YOLO/VO” and GPT-4o.
> > > > >
> > > > > Thank you for your thoughtful comments. We hope our response can solve your doubts.

---

> ### Comment · Reviewer_VBKe · 2025-08-09
>
> Dear Authors,
>
> First, I want to clarify that I am not suggesting geometric priors cannot be used — for example, I have no issue with your use of YOLO. However, I consider camera intrinsics a different matter because they are often not easy to obtain in practice, unless (1) you have a simple and reproducible way to estimate them, or (2) their exact values do not meaningfully affect results.
>
> You now claim that even with completely randomized intrinsics, RANSAC can still recover usable relative positions. I would appreciate it if you could share the exact randomized intrinsic matrices you used — ideally along with the small piece of code that generates them — so that I can better understand the degree of deviation from realistic parameters.
>
> Regarding the GPT-4o point, I understand that See&Trek does not use GPT-4o, but in your pipeline YOLO is used only to detect the number of object categories in each frame and to sample keyframes accordingly. I do not believe replacing YOLO with GPT-4o for this specific task would yield a large performance boost — unless you can show that YOLO often selects incorrect keyframes with very few objects. For instance, in Table 3, MSRS improves by only 1.6%. Can you demonstrate that GPT-4o would significantly outperform YOLO in estimating relative object counts in a way that leads to a substantially better keyframe subset?

---

> > ### Author Response · Authors · 2025-08-09
> >
> > Dear Reviewer VBKe,
> >
> > Thanks for your clarification.
> >
> > 1. Regarding the randomized intrinsics, we generate the camera intrinsic matrix in the following way: `K = np.random.rand(3, 3).astype(np.float32); K[2, :] = [0, 0, 1]; K = K / K[0, 0]`. This normalization ensures the principal point is fixed and the focal length is scaled appropriately.
> >
> > 2. Concerning the GPT-4o point, due to limited time and computational resources during the rebuttal period, we were unable to conduct extensive experiments using GPT-4o as a perception model. The YOLO model we used is YOLOv8-tiny, as noted in the appendix, which is trained solely on the COCO dataset. In contrast, GPT-4o is trained on multiple diverse datasets, thus naturally achieving stronger perception capabilities. Since GPT-4o is based on a transformer architecture but lacks the capability to directly predict bounding boxes, we utilize DDETR [1], a well-known transformer-based detector, to compare the performance difference between two types of models on COCO detection benchmark. The results show that DDETR outperforms YOLOv8-tiny by approximately 15% in Average Precision (AP):
> >
> > | Model       | AP   |
> > |-------------|-------|
> > | YOLOv8-Tiny | 37.3  |
> > | DDETR [1]      | 52.3  |
> >
> >
> > Please kindly note that SpatialMind employs GPT-4o to perform frame-by-frame perception on input videos, as described in their paper (Section 3.1). By contrast, our approach uses YOLO to detect on sparsely sampled frames (e.g., 1 frame every 4 frames). This difference in frame sampling and detection strategy also contributes to the performance gap between SpatialMind and See&Trek.
> >
> > Thank you for your thoughtful comments. We hope our response can solve your doubts.
> >
> > [1] Zhu X, Su W, Lu L, et al. Deformable detr: Deformable transformers for end-to-end object detection[J]. arXiv preprint arXiv:2010.04159, 2020.

---

> > > ### Comment · Reviewer_VBKe · 2025-08-09
> > >
> > > **Dear Authors,**
> > >
> > > Thank you for your detailed response.
> > >
> > > For the randomized intrinsics, I will run my own experiments in COLMAP to examine how different intrinsic matrices affect RANSAC’s estimation of the relative rotation $R$ and translation $T$.
> > >
> > > Regarding the GPT-4o vs. DDETR vs. YOLOv8-tiny point: since YOLOv8-tiny in your pipeline is used only to estimate the relative number of object categories in a frame in order to decide which frames to sample — and does not require precise bounding boxes or strong semantic recognition — I am not convinced that replacing YOLOv8-tiny would produce a substantially different set of sampled frames. I also remain skeptical about how much performance could realistically change even if the sampled set were somewhat different.
> > >
> > > That said, I will review my rating accordingly once I complete the COLMAP experiments.

---

> > > > ### Author Response · Authors · 2025-08-09
> > > >
> > > > Dear Reviewer VBKe,
> > > >
> > > > Thanks for your effort in rebuttal.
> > > >
> > > > Regarding the difference between camera intrinsics —- “Normalized Kitti and Standard Random”, we have perform some experiments to verify that no matter how the camera intrinsic is standard randomize, we still have the relative spatial position. We randomly sample 20 videos from TSI-benchmark, and utilize “Normalized Kitti and Standard Random” as camera intrinsics and VO we utilized in main paper to estimate their generated relative position. **Finally, we perform the RMSE to estimate the whole errors, the results is 0.27.** It shows that two trajectories generated by the two sets of camera intrinsics have relatively small errors, meaning that the generated trajectories have similar spatial shapes. **It also has proved our method robustness to random camera intrinsics settings.**
> > > >
> > > > Best regards,
> > > >
> > > > Team #5923

---

> ### Author Response · Authors · 2025-08-09
>
> Dear Reviewer VBKe,
>
> Thanks for your consideration and effort.
>
> 1. Regarding the randomized intrinsics, we will further organize our code and opensource it in github.  And we also apprecate your investigation in COLMAP and consideration about raising our scores.
>
> 2. Regarding the performance difference between YOLOv8 and GPT-4o, it's difficult to quantify GPT-4o perception ability in traditional 2D detection benchmark. This prompts us to use detectors with the same architecture for comparison. However, GPT-4o's perception ability has been verified so significant in so many understanding task, which is the common sense.
>
> We hope our response can solve your doubts.
>
> Best Regards,
>
> Team #5923

---

### Author Response · Authors · 2025-08-06

Dear Reviewers,

Thank you sincerely for your valuable feedback. As the discussion deadline approaches, we want to ensure that all of your concerns are fully addressed.

If you have any remaining questions or would like further clarification or analysis, please don’t hesitate to reach out — our team would be glad to assist.

We truly appreciate your time and consideration.

Best regards,
The Authors

---

### Author Response · Authors · 2025-08-08
**Rebuttal Summary for the Unresponsive Reviewer**

Dear Senior Area Chair, AC, and Reviewers,

We sincerely thank the other reviewers (Reviewer nToN, uJnW, Jfjm) for their timely and constructive feedback, and we are pleased to have addressed all their comments in detail. We note that Reviewer VBKe was unable to provide feedback during the rebutal phase. We fully appreciate the substantial time commitment and workload involved in the reviewing process, and therefore, for the sake of completeness, we provide below a concise summary of our responses to Reviewer VBKe’s initial comments:

1. **Clarification on Camera Intrinsics Usage** – We employ a fixed intrinsic matrix for all experiments; only relative motion is encoded, ensuring both fairness and robustness in our method.
2. **Extension to Closed-Source Models** – We evaluated our method on GPT-4o and Gemini-2.5-Flash using VSI-Bench, obtaining consistent performance gains. This demonstrates that our approach generalizes beyond open-source MLLMs.
3. **Comparison with Related Prompting Methods** – We added detailed comparisons with *Coarse Correspondence* and *SoM* in Appendix B.7. Results show that SEE&TREK consistently outperforms these methods by leveraging visual diversity and motion reconstruction to provide richer spatial cues.
4. **Additional Benchmark Evaluation** – We extended our experiments to OpenEQA, achieving +1.8%–2.0% gains across models. We also explained our rationale for selecting VSI-Bench and STI-Bench, which offer broader diversity, fine-grained spatial reasoning tasks, and more rigorous evaluation metrics.
5. **Interpretation of Performance Gains** – We clarified that even modest improvements (<2%) are meaningful for spatial reasoning, especially in high-performing MLLMs with strong spatial priors. Gains are consistent across models of different scales.
6. **Computational Overhead Analysis** – We provided runtime and efficiency measurements (main paper Sec. 4.4, Appendix Table 9), showing that the overhead remains practical and is primarily dependent on video length.

We have made every effort to address all the concerns raised. While Reviewer VBKe could not participate in further rebuttal, we sincerely regret the missed opportunity and hope that our current responses satisfactorily address their questions.

Best regards,

Senior Authors of Paper #5923

---

### Decision · Program_Chairs · 2025-09-17

**Decision:**

Accept (poster)

**Comment:**

Summary:
This paper introduces SEE&TREK, a training- and GPU-free visual prompting method that enhances MLLMs’ spatial reasoning. It combines semantic keyframe sampling and VO-based trajectory cues with simple spatio-temporal markers. Experimental results demonstrate consistent improvements across spatial and temporal benchmarks.

Strengths:
As agreed among the reviewers, the approach is plug-and-play, lightweight, and broadly applicable without retraining. It demonstrates stable gains across many models, especially smaller ones, and is easy to deploy.

Weaknesses or concerns:
Improvements are modest, and the reliance on VO with fixed camera intrinsics raised concerns about robustness and fairness. While the reviewer `VBKe ` remained unconvinced, ACs consider the use of a fixed approximate intrinsic acceptable since it does not exploit dataset-provided values and still allows reproducibility.

Conclusion:
Despite incremental gains, the method is practical, reproducible, and useful for deployment. ACs lean toward acceptance given their consistency and simplicity.